# Metabolism of L-arabinose converges with virulence regulation to promote enteric pathogen fitness

Curtis Cottam [1], Rhys T. White [2,3], Lauren C. Beck [4], Christopher J. Stewart [4], Scott A. Beatson [3], Elisabeth C. Lowe[1], Rhys Grinter [5] & James P. R. Connolly [1] ✉

Virulence and metabolism are often interlinked to control the expression of essential colonisation factors in response to host-associated signals. Here, we identified an uncharacterised transporter of the dietary monosaccharide L-arabinose that is widely encoded by the zoonotic pathogen enterohaemorrhagic *Escherichia coli* (EHEC), required for full competitive fitness in the mouse gut and highly expressed during human infection. Discovery of this transporter suggested that EHEC strains have an enhanced ability to scavenge L-arabinose and therefore prompted us to investigate the impact of this nutrient on pathogenesis. Accordingly, we discovered that L-arabinose enhances expression of the EHEC type 3 secretion system, increasing its ability to colonise host cells, and that the underlying mechanism is dependent on products of its catabolism rather than the sensing of L-arabinose as a signal. Furthermore, using the murine pathogen *Citrobacter rodentium*, we show that L-arabinose metabolism provides a fitness benefit during infection via virulence factor regulation, as opposed to supporting pathogen growth. Finally, we show that this mechanism is not restricted to L-arabinose and extends to other pentose sugars with a similar metabolic fate. This work highlights the importance integrating central metabolism with virulence regulation in order to maximise competitive fitness of enteric pathogens within the host-niche.

The mammalian gastrointestinal tract poses a formidable barrier to infection by foreign pathogens. Invaders must sidestep a combination of intrinsic host defences as well as overcome colonisation resistance by the native gut microbiota[1]. Accordingly, bacterial pathogens have evolved many unique strategies to effectively compete with the microbiota and cause infection within a favourable host-niche[2]. This includes a combination of pathogen-specific virulence mechanisms and metabolic adaptations that increase within-host fitness. Importantly, virulence and fitness factor regulation often coincide and are dynamically controlled in response to the host environment to maximise competitiveness therein[3–5].

Enterohaemorrhagic *Escherichia coli* (EHEC) is a zoonotic pathogen that is carried asymptomatically by ruminant mammals and transmitted to human hosts typically via contaminated meat and fresh produce. In humans, EHEC causes severe diarrhoeal illness and, in extreme cases, renal failure[6,7]. EHEC is a member of the attaching and effacing (A/E) family of pathogens, which intimately colonise the colonic epithelium forming characteristic pedestal-like lesions on the

[1]Newcastle University Biosciences Institute, Newcastle University, NE2 4HH Newcastle-upon-Tyne, UK. [2]Institute of Environmental Science and Research, Wellington, New Zealand. [3]Australian Infectious Diseases Research Centre and School of Chemistry and Molecular Biosciences, The University of Queensland, Brisbane, QLD, Australia. [4]Newcastle University Translation and Clinical Research Institute, Newcastle University, NE2 4HH Newcastle-upon-Tyne, UK. [5]Department of Microbiology, Biomedicine Discovery Institute, Monash University, Clayton, VIC, Australia. ✉e-mail: James.Connolly2@newcastle.ac.uk

surface of host cells[8,9]. A/E pathogenesis is defined by the activity of a type 3 secretion system (T3SS) encoded on a ~35 kb horizontally acquired island known as the locus of enterocyte effacement (LEE)[10–12]. This T3SS translocates more than 30 effector proteins - encoded on the LEE and several additional horizontally acquired elements, termed O-islands (OIs) - that collectively subvert host-cell function[13–16]. The murine pathogen *Citrobacter rodentium* also encodes the LEE and has been adopted as the relevant surrogate model to study EHEC pathogenesis in vivo due to its dependency on the T3SS, comparable pathology and lack of requirement to pre-treat mice with antibiotics[17–19].

The LEE-encoded T3SS is essential for overcoming microbiota-dependent colonisation resistance[20,21]. Epithelial attachment does not depend on a tissue receptor interaction. Instead, the LEE is uniquely controlled in response to a multitude of host and microbiota-derived signals such as sugars, amino acids, short- and long-chain fatty acids and hormones that are integrated into a regulatory network ensuring correct spatial deployment of the T3SS[3–5]. For example, D-glucose represses the LEE whereas gluconeogenic substrates encountered at the epithelial surface (such as succinate) enhance its expression[22,23]; L-arginine found in abundance in the gut is directly sensed to activate the LEE, whereas the amino acid D-serine, abundant in the urinary tract, represses the LEE thus restricting EHEC to the gut niche[24–26]. A/E pathogenesis therefore hinges on nutrient availability, by promoting competition with the commensal microbiota as well as regulating essential virulence factors.

*E. coli* as a species relies on cross-feeding of microbiota-liberated mono- and disaccharide sugars as it cannot degrade complex dietary polysaccharides[27]. However, EHEC has evolved to utilise a unique hierarchy of sugars compared to commensal *E. coli*, therefore implying a strategy to limit competition with similar members of the microbiota for critical resources[28]. The pentose monosaccharide L-arabinose is one such resource and is highly abundant in nature as a major constituent of plant cell walls[29]. *E. coli* transports L-arabinose into the cell via the H+ symporter AraE and the ATP-binding cassette (ABC) transporter AraFGH. The transcription factor AraC then senses cytosolic L-arabinose and upregulates both transporters as well as the AraBAD enzymes essential for its metabolism, a mechanism enhanced by the presence of cyclic AMP that increases during D-glucose depletion[30]. Here, we describe the identification of a previously unknown L-arabinose uptake system that is widely encoded by T3SS-encoding EHEC strains in nature and we hypothesise that this system gives EHEC an enhanced capacity to scavenge L-arabinose in the gut. Guided by this discovery, we show that L-arabinose utilisation is exploited by A/E pathogens during infection. Surprisingly, this occurs independently of its role as a source of nutrition. Instead, this advantage is achieved via a mechanism that links metabolism and virulence regulation. Transcriptome analysis revealed that L-arabinose upregulates expression of the LEE-encoded T3SS, which relies on the uptake and breakdown of L-arabinose in the cell acting as a regulatory trigger, as opposed to supporting pathogen growth in vivo. Finally, we provide evidence that this mechanism is also triggered by the metabolism of other pentose sugars that share a similar metabolic fate in the cell. Our results therefore highlight how nutrient utilisation can provide a fitness advantage via the convergence of virulence regulation and metabolism.

## Results

### An accessory L-arabinose uptake system is widely encoded by EHEC isolates in nature

While analysing our previously determined *C. rodentium* in vivo transcriptome, we noticed that genes related to monosaccharide uptake and metabolism were some of the most highly upregulated during infection[21]. Among these was ROD_24811, an uncharacterised gene displaying ~20-fold induction in the colon and predicted to encode a periplasmic binding protein of a simple sugar ABC transporter. A search for homologues in the prototypical EHEC strain TUV93-0 did not find an exact match but instead identified a similar uncharacterised 4.4 Kb ABC-transporter locus (>60% nucleotide identity) that appeared to be EHEC-specific, displaying a unique genetic context, being located on OI-island 17. Importantly, this system was previously identified as being highly expressed during human infection and during the growth of EHEC in bovine intestinal contents, but its function was unknown[31,32]. We, therefore, focused our efforts on the role of this system given its relevance to EHEC and potential interaction with the ruminant and human hosts. This locus was predicted to encode a periplasmic binding protein (Z0415), an ATPase (Z0416-7) and two permease subunits comprised of α-helices (Z0418/Z0419), characteristic of Type II ABC transporters[33]. In silico modelling of Z0415-9 using AlphaFold2 supported this, by displaying the expected modular structure of an inner membrane ABC transporter (Supplementary Fig. 1). InterPro analysis of the associated amino acid sequences identified domain signatures related to pentose monosaccharide substrate specificity and comparison with known *E. coli* systems supported that Z0415 clustered closely with monosaccharide ABC transporters (Supplementary Fig. 2). Presence/absence analysis of Z0415-9 carriage amongst 949 representative *E. coli* genomes revealed that the locus is not completely conserved across the species phylogeny (Fig. 1). For instance, Z0415-19 is carried predominantly amongst isolates from phylogroups B1 and E (397/445 strains), comprised largely of EHEC strains, whereas phylogroups A and B2 (largely commensal and extraintestinal isolates, respectively) largely lack this system. Strikingly, there was a significant correlation between Z0415-9 and LEE carriage (Odds ratio = 35.9; *P* < 0.001 Fisher's Exact test), with the converse scenario (LEE positive, Z0415-9 negative) being an incredibly rare event (8/948 strains) (Fig. 1c). This suggests that there may be a related evolutionary pressure for LEE-encoding EHEC strains to acquire or retain Z0415-9, implicating its associated function in pentose sugar scavenging as being potentially beneficial for EHEC infection.

To assess the function of Z0415-9, we generated a transcriptional reporter (EHEC transformed with pMK1lux-P$_{Z0415-9}$) and cultured this in MEM-HEPES supplemented with either D-xylose, D-ribose or L-arabinose (Supplementary Fig. 3a). Z0415-9 promoter activity was induced in a concentration-dependent manner exclusively in response to L-arabinose and we validated this result using RT-qPCR (Fig. 2a; Supplementary Fig. 3b, c). Additionally, we confirmed that Z0415-9 induction occurs exclusively in response to L-arabinose and not D-arabinose, the less common form of this sugar found in nature (Supplementary Fig. 3d). We next hypothesised that Z0415-9 might be regulated similarly to the canonical L-arabinose machinery. The activity of pMK1lux-P$_{Z0415-9}$ in a ΔaraC background was completely abolished during growth with L-arabinose (Fig. 2b). Importantly this could be complemented by expressing araC in trans, suggesting that this horizontally acquired system is co-regulated with the canonical L-arabinose utilisation machinery (Fig. 2c). We, therefore, named this locus Aau, for accessory L-arabinose uptake system.

To study the role of Aau in EHEC growth and fitness, we employed strain ZAP193 (ST11) encoding a functionally intact Aau locus, due to TUV93-0 containing a SNP in the ATPase component of the transporter that results in a premature stop codon at position 259. Deletion of Aau in ZAP193 did not result in a significant defect during in vitro growth on L-arabinose as a sole carbon source, likely due to the presence of the canonical transporters AraE or AraFGH (Supplementary Fig. 4). As an alternative way of assessing if Aau could potentially provide a benefit to EHEC, we cloned the entire locus from ZAP193 into plasmid pSU-PROM (pSU-*aau*) where it is under constitutive control of the Tat promoter. Growth of TUV93-0 was first compared to ΔaraE, a mutant which displays a major fitness defect on L-arabinose as the sole carbon source (Supplementary Fig. 5a/b). However, when pSU-*aau* was

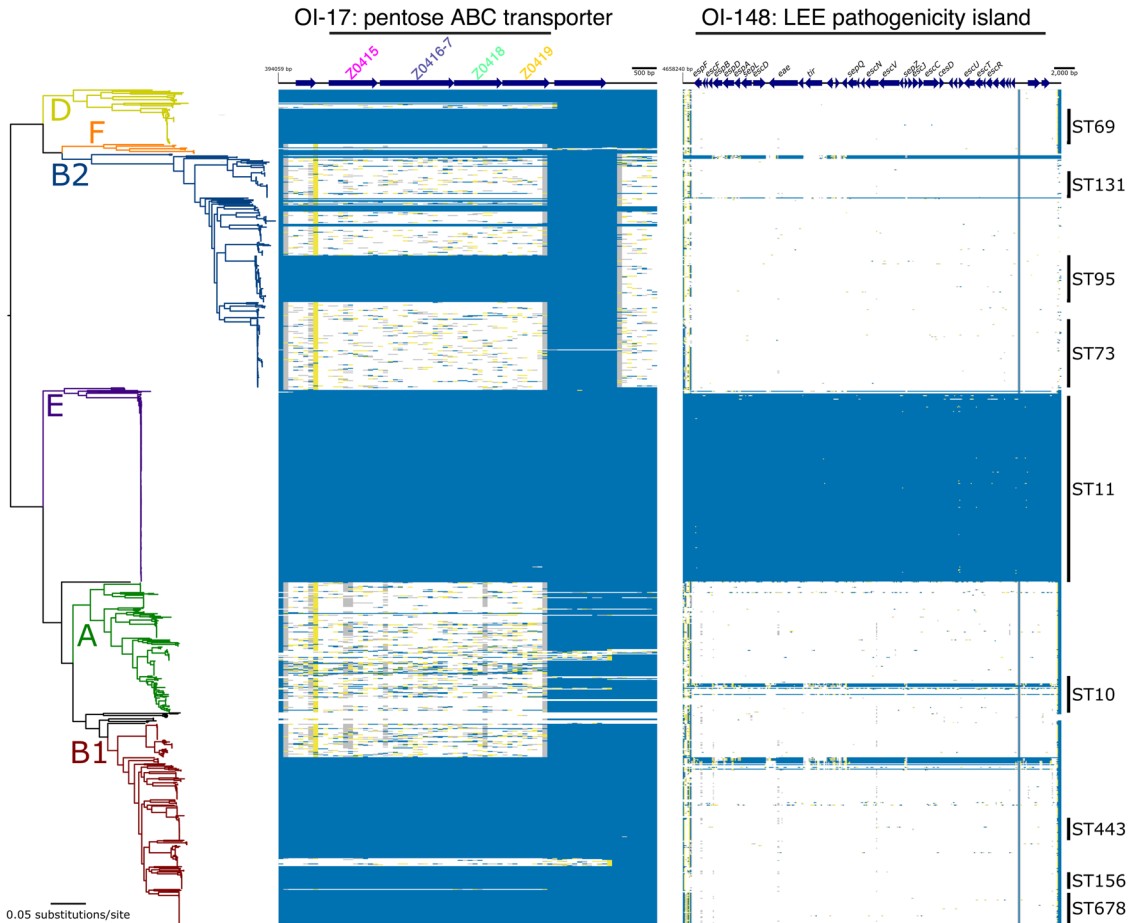

**Fig. 1 | A horizontally acquired pentose sugar ABC transporter widely encoded by enteric EHEC strains.** Maximum likelihood analysis depicting the core-genome phylogeny of 949 *E. coli* isolates built from 245,518 core-genome single-nucleotide polymorphisms called against the reference chromosome EDL933. Phylogeny is rooted according to the actual root by *Escherichia fergusonii* ATCC 35469, which has been omitted for visualisation. Branch colours indicate the six main phylogenetic groups. Branch lengths and scale bar represent the number of nucleotide substitutions per site. The presence/absence of Z0415-9 or the LEE is based on the uniform coverage at each 100 bp window size. Coverage is shown as a heat map where ≥80% of identity is highlighted in blue, ≥50% of identity is highlighted in yellow, and ≥1% is highlighted in grey. White plots indicate absent regions.

introduced into the Δ*araE* background we observed that growth was partially recovered (Supplementary Fig. 5c). This suggests that Aau has the capacity to enhance ʟ-arabinose uptake in combination with existing transporters and thus potentially increase fitness during ʟ-arabinose utilisation.

We next hypothesised that Aau may play a more important role in a more relevant host-context during nutrient competition. To test this, we orally inoculated streptomycin-treated BALB/c mice with a 1:1 mixture of wild-type EHEC and Δ*aau* (Fig. 2d)[28]. While both strains initially colonised equally well, Δ*aau* was significantly attenuated for longer-term persistence displaying a 10–100-fold decrease in competitive fitness from day 9 onwards (Fig. 2e). This result implies that Aau provides a fitness benefit to EHEC within the complex and dynamic gut niche.

### ʟ-arabinose alters the EHEC transcriptome and enhances expression of the LEE-encoded T3SS

Our discovery of Aau and its positive regulation in response to ʟ-arabinose prompted us to investigate if other EHEC genes exhibit enhanced expression in response to this sugar. Nutrients often act as signals that modulate virulence gene expression. We, therefore, hypothesised that ʟ-arabinose may affect the expression of key EHEC virulence genes, as well as being able to act as a source of nutrition. To test this, we performed RNA-seq on EHEC cultured in MEM-HEPES with and without 1 mg/ml ʟ-arabinose to late exponential phase (~5 h).

Strikingly, we identified 1187 significant (>1.5 fold change; FDR $P < 0.05$) differentially expressed genes (DEGs) (Fig. 3a; Supplementary Data 1). As expected, the ʟ-arabinose utilisation genes (*araBAD*), all known transporters (*araE, araFGH, araJ* and *ytfQRTyjfF*), as well as genes encoding Aau were among the most strongly upregulated displaying increased expression of up to 42-fold ($P < 0.001$)[30,34]. Functional network analyses identified that the large abundance of DEGs unrelated to ʟ-arabinose clustered into diverse biological pathways including metabolism, quorum sensing, signalling and regulation of biofilm formation (Supplementary Fig. 6a–d). While we anticipated shifts in the expression of genes related to metabolism, we also noticed that the majority of the LEE pathogenicity island displayed increased expression, with several genes (*escL, escT, espG* and *map*) reaching statistical significance (Fig. 3b). Additionally, the non-LEE-encoded effectors *nleF* and *nleG6-3* were also significantly upregulated in response to ʟ-arabinose. This data suggests that exposure to ʟ-arabinose may influence the expression of virulence genes that are not related to pentose utilisation.

The LEE is essential for EHEC pathogenicity and is responsive to gut-associated cues. Its associated genes are encoded largely across five polycistronic operons (LEE1-5) and are co-regulated by the activity of the master regulator Ler (encoded by the first open reading frame of LEE1)[12]. We, therefore, reasoned that ʟ-arabinose may provide a fitness advantage through regulating T3SS activity either in concert with or independent from a role in the host-associated nutritional

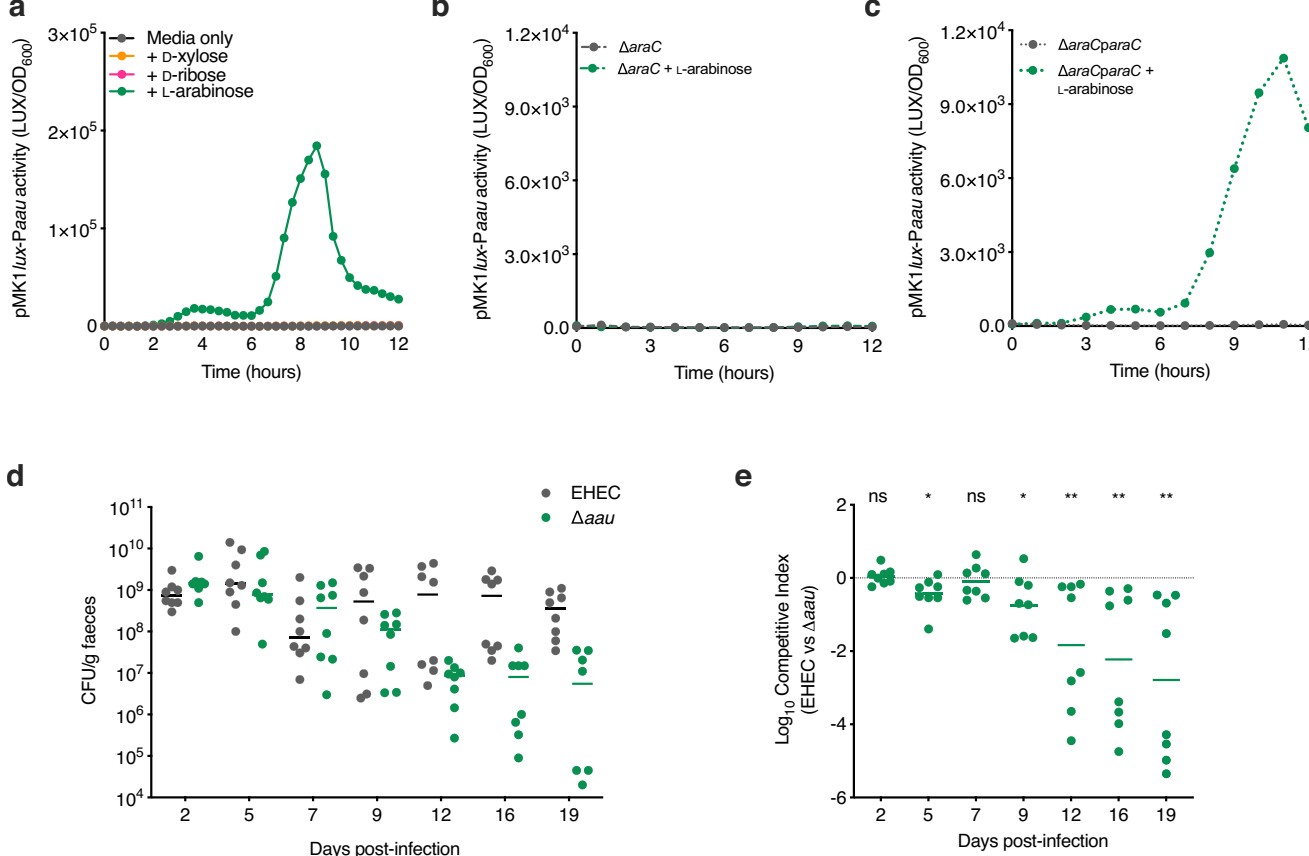

**Fig. 2 | Aau is induced by ʟ-arabinose and required for EHEC fitness in the mouse gut. a** Transcriptional reporter assay of EHEC transformed with pMK1*lux*-P*aau* cultured in MEM-HEPES alone or supplemented with 0.5 mg/ml ʟ-arabinose (green), ᴅ-ribose (pink) or ᴅ-xylose (orange). Data are depicted as luminescence units (LUX) divided by the optical density ($OD_{600}$) of the culture at each timepoint. **b** pMK1*lux*-P*aau* reporter assay utilising wild type (WT) EHEC and the Δ*araC* grown in MEM-HEPES alone (grey) or supplemented with ʟ-arabinose (green). **c** pMK1*lux*-P*aau* reporter assay utilising Δ*araC* and Δ*araC* + p*araC* grown in MEM-HEPES alone or supplemented with ʟ-arabinose. Graphs in (**a**–**c**) are representative of three

independent repeats. **d** Faecal shedding dynamics of Streptomycin-treated BALB/c mice (*n* = 8) colonised with a 1:1 mixture of EHEC (grey) and Δ*aau* (green). **e** Competitive index of EHEC versus Δ*aau* during colonisation of Streptomycin-treated BALB/c mice (*n* = 8). Data points indicate the fold decrease in Δ*aau* CFU recovered per faecal sample in comparison to wild-type EHEC. Statistical significance was determined by a two-tailed Wilcoxon signed-rank test (significant *P* = 0.0156, 0.0391, 0.0078, 0.0078 and 0.0078 where indicated from left to right). *, ** and ns indicate *P* < 0.05, *P* < 0.01 or not significant respectively. Source data are provided in the Source Data file.

competition. To assess the dynamics of LEE induction by ʟ-arabinose, we engineered a transcriptional reporter of the LEE1 promoter, used as a proxy to measure T3SS expression (EHEC transformed with pMK1*lux*-P*LEE1*). Culture of this strain in MEM-HEPES (LEE-inducing conditions) supplemented with various concentrations of ʟ-arabinose resulted in no increase in growth rate but a significant and prolonged increase in LEE promoter activity from late exponential phase onwards (Fig. 3c). RT-qPCR analysis confirmed a significant increase in transcription across all five LEE operons only at this later stage of growth, thus explaining the moderate increase observed in our RNA-seq analysis (Supplementary Fig. 7a). We detected an increase in LEE expression in the presence of ʟ-arabinose at concentrations as low as 50 μg/ml (Supplementary Fig. 7b), within range of the amount quantified from the luminal content and faeces of mice maintained on a conventional diet (Supplementary Fig. 7c). We next observed an increased accumulation of T3SS associated effector proteins in the cytosol and cell-free supernatant by SDS-PAGE and western blot analysis (Fig. 3d). To test if this increase in T3SS expression and function resulted in enhanced host-cell interaction, we quantified adhesion of EHEC to cultured epithelial cells by fluorescence microscopy (Fig. 3e). Pre-exposing EHEC to ʟ-arabinose before infection of epithelial cells significantly increased the number of attached EHEC and associated A/E lesions (identified as foci of actin accumulation) per infected cell (Fig. 3f). This was in parallel with a >10% increase in the number of

individually infected cells in the presence of ʟ-arabinose. These results collectively show that ʟ-arabinose enhances the expression and function of the LEE-encoded T3SS, resulting in a greater capacity to cause A/E lesions on the host epithelial surface.

## Metabolism of ʟ-arabinose is essential to enhance T3SS expression

Nutrient sensing typically occurs either from the environmental (via two-component systems) or via a cognate transcription factor in the cytosol[35]. Due to ʟ-arabinose utilisation occurring at three levels (uptake via multiple transporters, sensing by a dedicated transcription factor and metabolism via AraBAD), this left the mechanism of T3SS regulation by ʟ-arabinose unclear, as in principle it could occur at any of these three stages (Supplementary Fig. 8a)[30]. We therefore utilised mutants in each component, allowing compartmentalisation of each stage to explore the mechanism of T3SS enhancement (Supplementary Fig. 8b). We first measured LEE activity in Δ*araC* and observed a complete reversal of ʟ-arabinose enhanced T3SS expression. This was complemented in trans suggesting that sensing via AraC may directly regulate LEE expression (Fig. 4a). Next, we measured LEE activity in the Δ*araBAD*, Δ*araFGH* and Δ*araE* backgrounds. This revealed that enhanced LEE expression was also abolished in Δ*araBAD*, but not in the Δ*araFGH* background and partially in Δ*araE*, likely since there are multiple routes for ʟ-arabinose uptake but in line with the major

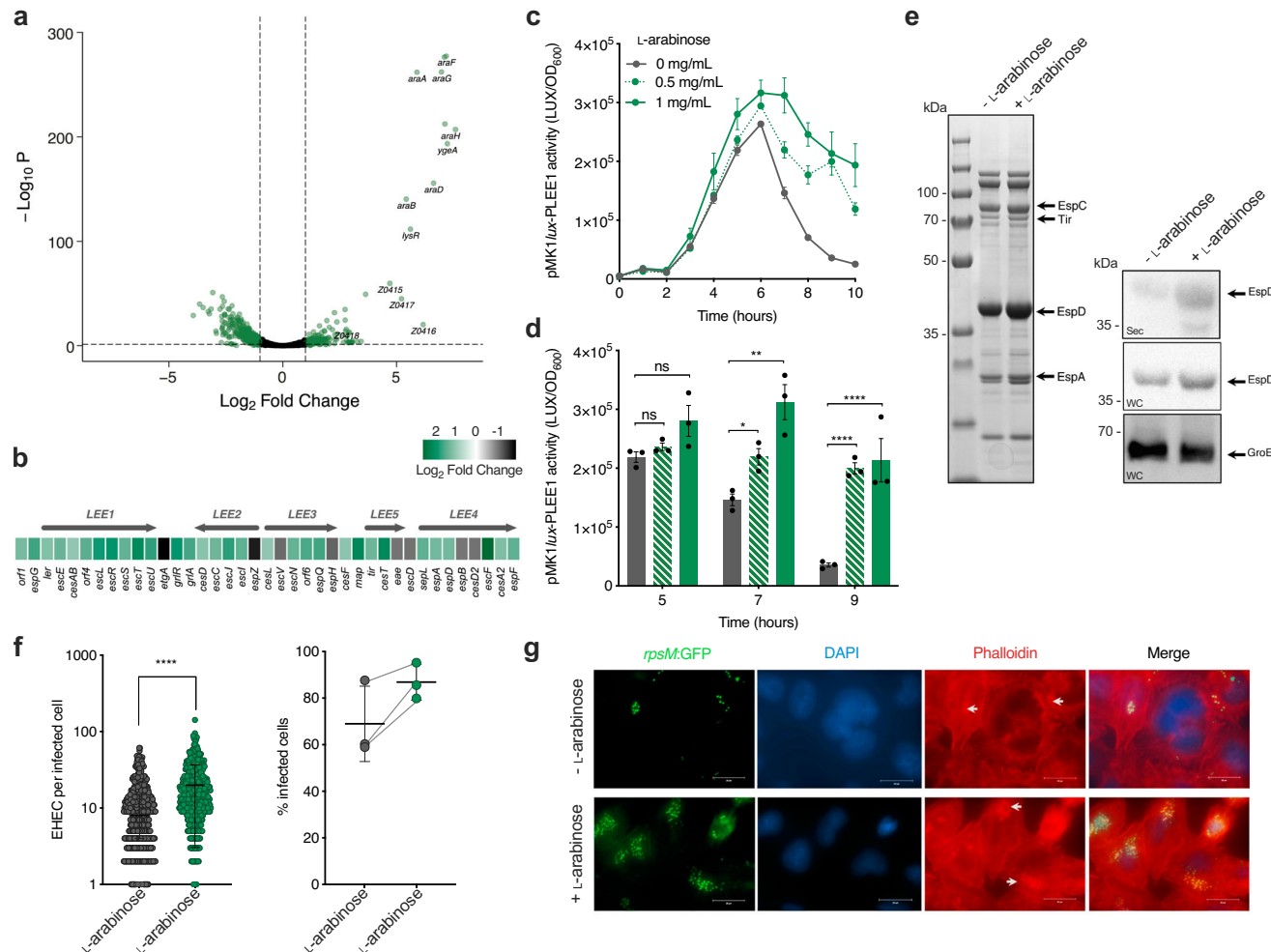

**Fig. 3 | L-arabinose enhances the expression and function of the LEE-encoded T3SS. a** Volcano plot depicting RNA-seq expression patterns of EHEC cultured in MEM-HEPES supplemented with L-arabinose versus media control. Significantly differentially expressed genes determined using DEseq2 (FDR-corrected $P \leq 0.05$) are green with select L-arabinose-related genes labelled. **b** Expression profile of the LEE island from (**a**). The scale bar indicates $\text{Log}_2$ fold change in expression in response to L-arabinose. **c** Transcriptional reporter assay of EHEC transformed with pMK1*lux*-PLEE1 cultured in MEM-HEPES (grey) or supplemented with 0.5 or 1 mg/ml of L-arabinose (green dashed or solid line respectively). Data are depicted as luminescence units (LUX) divided by optical density ($\text{OD}_{600}$) at each timepoint. **d** Reporter assay statistical analysis at selected timepoints determined by two-tailed students' *t*-test (significant $P = 0.0129, 0.0061, 0.0001$ and 0.0086 where indicated from left to right). Error bars represent the standard error of the mean ($n = 3$ biological replicates). **e** SDS-PAGE analysis of secreted proteins identified from cell-free supernatant of EHEC culture in MEM-HEPES alone or supplemented with L-arabinose. The identity of known T3SS-associated effector proteins is labelled with arrows. Corresponding immunoblot analysis of EspD (LEE-encoded) or GroEL levels identified in secreted fractions (Sec) in the cell-free supernatant or whole-cell pellet (WC). Data is representative of three biological repeats. **f** Data derived from widefield fluorescence microscopy analysis of HeLa cells infected with EHEC cultured in MEM-HEPES alone ($n = 531$) or supplemented with L-arabinose (green, $n = 522$). Quantification of the number of EHEC lesions per infected HeLa cell was determined from collated data of 20 random fields of view. Statistical significance was determined using a two-tailed Mann–Whitney *U*-test ($P < 0.0001$). The graph on the right shows the average percentage of infected HeLa cells determined from three independent experiments. Error bars represent standard deviation. **g** Representative images of HeLa cells infected with EHEC cultured with and without L-arabinose. Channels are labelled and white arrows indicate A/E lesions as areas of condensed actin co-localised with EHEC cells. The scale bars indicate µM. Biological replicates were performed on three independent occasions with similar results. *, **, **** and ns indicate $P < 0.05$, $P < 0.01$, $P < 0.0001$ or not significant respectively for all graphs. Source data are provided in the Source Data file.

dependence on AraE for growth in vitro on this sugar (Fig. 4b; Supplementary Fig. 8b). In addition to this, we tested LEE expression in the Δ*aau* background showing that, at least in vitro, this system does not significantly mediate the enhanced LEE phenotype (Supplementary Fig. 9). To narrow down the mechanism further, we measured LEE expression in Δ*araC* transformed with plasmids constitutively expressing either *araBAD* and *araE* in parallel (pSU-*araBAD/E*) or *araE* alone (pSU-*araE*), allowing functional differentiation between uptake and downstream metabolism in the cytosol (Supplementary Fig. 8b). The rationale behind this experiment was that *araBAD* and *araE* expression (and therefore uptake and metabolism) is completely dependent upon AraC[36]. Therefore, bypassing the role of AraC using

constitutively expressed *araBAD/araE* allowed us to conclusively determine whether L-arabinose sensing by AraC, uptake or metabolism caused enhanced LEE expression. Complementation of Δ*araC* with pSU-*araBAD/E* resulted in complete restoration of L-arabinose enhanced LEE expression above wild-type levels, whereas constitutive expression of AraE alone (pSU-*araE*) had no restorative effect (Fig. 4c). Taken together, these data comprehensively show that L-arabinose metabolism (via AraBAD) in the cytosol is essential to enhance T3SS expression in response to this sugar.

To understand the ecological context of these results, we measured the temporal dynamics of *araBAD* transcription compared to the LEE, showing that induction of L-arabinose metabolism under these

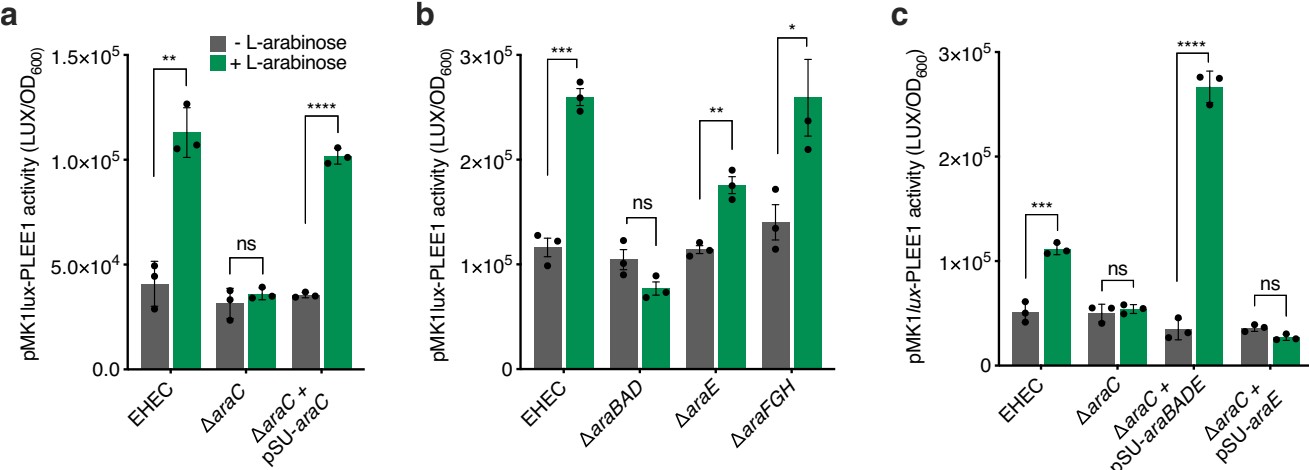

**Fig. 4 | L-arabinose metabolism is required to enhance T3SS expression.**
**a** Transcriptional reporter assay of wild-type EHEC, ΔaraC and ΔaraC + pSUaraC transformed with pMK1lux-PLEE1 cultured in MEM-HEPES alone (grey) or supplemented with L-arabinose (green). Data are depicted as luminescence units (LUX) divided by the optical density ($OD_{600}$) of the sample. All samples were taken at the same point in growth (hour 7). **b** pMK1lux-PLEE1 reporter assays performed in the EHEC, ΔaraBAD, ΔaraE and ΔaraFGH backgrounds. **c** pMK1lux-PLEE1 reporter

assays performed in the EHEC and ΔaraC backgrounds. For complementation, ΔaraC was transformed with either pSU-araBADE or pSU-araE. Statistical significance was determined using a two-tailed student's t-test (significant $P = 0.0014$, 0.0001, 0.0003, 0.0023, 0.0417, 0.0007 and 0.0001 where indicated from left to right on **a**–**c**). Error bars represent the standard error of the mean ($n = 3$ biological replicates). \*, \*\*, \*\*\*, \*\*\*\* and ns indicate $P < 0.05$, $P < 0.01$, $P < 0.001$, $P < 0.0001$ or not significant respectively. Source data are provided in the Source Data file.

conditions significantly correlates ($R = 0.92$; $P < 0.001$) with LEE expression dynamics (Supplementary Fig. 10a). MEM-HEPES contains D-glucose, which suppresses L-arabinose utilisation genes by catabolite repression[30,36]. Glycolytic growth is also known to reduce maximal LEE expression[22]. We therefore reasoned that the temporal dynamics of our mechanism likely represented the switch to L-arabinose catabolism as D-glucose became depleted from the medium. Using thin-layer chromatography, we measured the relative abundance of monosaccharides in cell-free supernatant of EHEC cultured in MEM-HEPES alone or supplemented with L-arabinose (Supplementary Fig. 10b). The data confirmed that D-glucose depletion coincided with activation of AraBAD (and as such, the LEE). This offered a logical explanation as to why enhanced LEE expression in response to L-arabinose metabolism was observed only at this stage of growth when the repressive effects of D-glucose on araBAD and the LEE would no longer be a factor (Supplementary Fig. 10c). This suggests that dynamic changes in nutrient availability that likely occur within the complex host environment could have important downstream effects on the mechanisms of virulence factor regulation.

### L-arabinose metabolism is required for maximal fitness during enteric infection

While L-arabinose metabolism has been reported to provide a fitness advantage to *E. coli* during colonisation of streptomycin-treated mice, this model does not reflect A/E pathogenesis and therefore the role of L-arabinose utilisation during infection was unknown[28]. To test this, we employed *C. rodentium* as a model A/E pathogen, first confirming that L-arabinose also enhanced its T3SS expression (Supplementary Fig. 11). We next compared the ability of wild-type *C. rodentium* and a ΔaraBAD derivative to colonise BALB/c mice ($n = 10$). During the early stages of infection, bacterial burden in faeces was similar between the two strains (days 1–9 post-infection). However, during the infectious peak at day 13 and onwards into the resolving phase, we observed a significantly more rapid clearance of ΔaraBAD from the mice when compared to the wild type (Fig. 5a). To directly test if L-arabinose utilisation conferred a competitive fitness advantage, we orally infected mice with a 1:1 mixture of wild type *C. rodentium* and ΔaraBAD ($n = 10$). The ΔaraBAD mutant was more significantly outcompeted by the wild type in a competitive infection, displaying an increasing

fitness defect from day 9 until termination at day 21 (Fig. 5b). To determine whether the fitness defect was associated with tissue adhered or luminal *C. rodentium*, we quantified the bacterial burden of colon sections from mono-infected mice that were cleared of the luminal content. In agreement with the faecal counts, colon-associated ΔaraBAD was recovered in significantly fewer numbers than wild-type *C. rodentium* suggesting that L-arabinose metabolism actively promotes colonisation of host-tissue (Fig. 5c).

Finally, we aimed to determine whether the observed fitness defect of ΔaraBAD was driven by the inability to utilise L-arabinose as a nutrient or lack of its input as a positive stimulus for regulation of the LEE. To test this, we generated ΔaraBAD in *C. rodentium*[Pler-const], a strain that expresses the LEE constitutively via a single base deletion in the -30 element of the *ler* promoter[21,37], and tested its colonisation dynamics in mice ($n = 9$). While *C. rodentium*[Pler-const] displayed similar colonisation dynamics to that of the wild-type strain (Supplementary Fig. 12), ΔaraBAD[Pler-const] no longer displayed a significant decrease in within-host fitness (Fig. 5d). This suggested that constitutive expression of the T3SS can overcome the fitness defect of ΔaraBAD by mimicking the stimulatory effect that L-arabinose metabolism imparts on the LEE and suggesting that during a natural infection *C. rodentium* does not require L-arabinose as a source of nutrition.

### An intrinsic, generalised mechanism of T3SS regulation via pentose sugar metabolism

L-arabinose metabolism, like other aldopentose sugars such as D-ribose and D-xylose, can generate cellular energy via the pentose phosphate shunt[30]. While each sugar can support *E. coli* growth using dedicated genes, their metabolism converges at the generation of D-xylulose-5-phosphate, which enters the Embden–Meyerhof pathway and ultimately produces pyruvate to generate cellular energy (Fig. 6a). It has been previously shown that exogenous pyruvate can enhance LEE expression[38]. We, therefore, reasoned that the observed effect of L-arabinose on the expression of the T3SS could be due to excess pyruvate generation and hypothesised that D-ribose or D-xylose may have a similar effect, given their common cellular fate. Strikingly, enhanced LEE expression dynamics were also observed when EHEC was cultured in the presence of either sugar (Fig. 6b). Importantly, deletion of the D-ribose utilisation genes (ΔrbsDACBKR) eliminated the

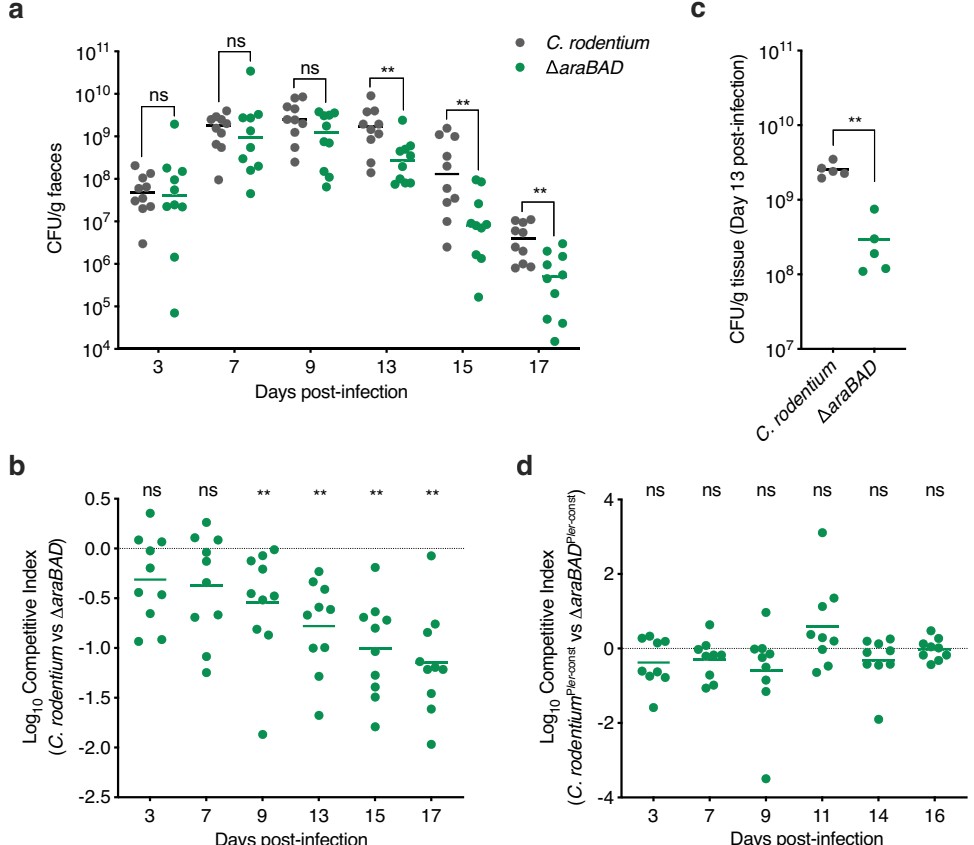

**Fig. 5 | L-arabinose metabolism promotes *C. rodentium* fitness in vivo by enhancing T3SS expression. a** Faecal shedding dynamics of BALB/c mice (*n* = 10) orally infected with either *C. rodentium* (grey) or Δ*araBAD* (green). Data points represent the CFU/ml for individual mice as determined from faecal pellets at each timepoint. Statistical significance was determined by a two-tailed Mann–Whitney *U*-test (significant *P* = 0.0129, 0.0061, 0.0001 and 0.0086 where indicated from left to right). **b** Competitive index of *C. rodentium* versus Δ*araBAD* during infection of BALB/c mice (*n* = 10). Mice were orally infected with a 1:1 mixture of both strains. Data points indicate the fold decrease in Δ*araBAD* CFU recovered per faecal sample in comparison to wild-type *C. rodentium*. Statistical significance was determined by

a two-tailed Wilcoxon signed-rank test (significant *P* = 0.0129, 0.0061, 0.0001 and 0.0086 where indicated from left to right). **c** CFU per gram of colon tissue from mice (*n* = 5) orally infected with either *C. rodentium* (grey) or Δ*araBAD* (green). Statistical significance was determined by a two-tailed Mann–Whitney *U*-test (*P* = 0.0129). **d** Competitive index of *C. rodentium*^Pler-const versus Δ*araBAD*^Pler-const during infection of BALB/c mice (*n* = 9). Mice were orally infected with a 1:1 mixture of both strains. Statistical significance was determined by a two-tailed Wilcoxon signed-rank test. ** and ns indicate *P* < 0.01 or not significant respectively for all graphs. Source data are provided in the Source Data file.

ability to grow on this sugar as a sole carbon source and, as predicted, abolished the observed increase in T3SS expression, suggesting that the common features of cellular pentose sugar metabolism determine the regulatory effect on the LEE (Fig. 6c). This is in line with our finding that L-arabinose must be metabolised to exert its T3SS regulatory effect. Finally, we confirmed that the addition of pyruvate to the medium enhanced LEE expression and demonstrated that growth in the presence of both exogenous pyruvate and L-arabinose displayed an additive effect, further enhancing T3SS expression levels (Fig. 6d). This suggests that the common cellular fate of pentose sugar metabolism and generation of key downstream metabolites is a generalised mechanism by which A/E pathogens can regulate virulence gene expression.

## Discussion

Successful cross-feeding on nutrients derived from the host, microbiota or diet is essential for bacterial pathogens to overcome colonisation resistance and replicate within their preferred host-niche[1,2]. Moreover, nutrients that support growth can often be "sensed" as stimuli that trigger virulence gene expression, therefore coupling virulence and metabolism to maximise fitness[2–5]. Here, we have found that the metabolism of L-arabinose by A/E pathogens promotes within-host fitness by generating central metabolites that converge with the

regulation of virulence. While L-arabinose supports the growth of EHEC and *C. rodentium* in vitro, it is not required as a source of nutrition during infection of mice. Instead, its catabolism to pyruvate stimulates expression of the LEE-encoded T3SS, a virulence factor essential for colonisation, which provides an advantage during host-colonisation. We suggest that this underlying mechanism revolving around the generation of central metabolic products that enhance LEE expression is generalisable for other pentose sugars with a similar cellular fate. We also identified a novel ABC transporter, termed Aau, that is selectively induced in response to L-arabinose exclusively, suggesting that EHEC has an enhanced ability to scavenge L-arabinose in the gut and maximise its competitiveness through the convergence of virulence and metabolism.

Freter's nutrient-niche hypothesis suggests that within the complex, multi-species environment of the gut, an invading pathogen must be able to utilise at least one limiting nutrient better than the commensals it competes with[27,39]. L-arabinose-containing polysaccharides are a major component of dietary fibre, with free L-arabinose being highly enriched in the colon of humans and animals[40,41]. As such, non-starch hydrolysis and degradation of L-arabinose-containing polysaccharides by saccharolytic members of the gut microbiota could liberate free L-arabinose that is subsequently exploited by species such as *E. coli* during the process of cross-feeding[42–46]. Indeed, research

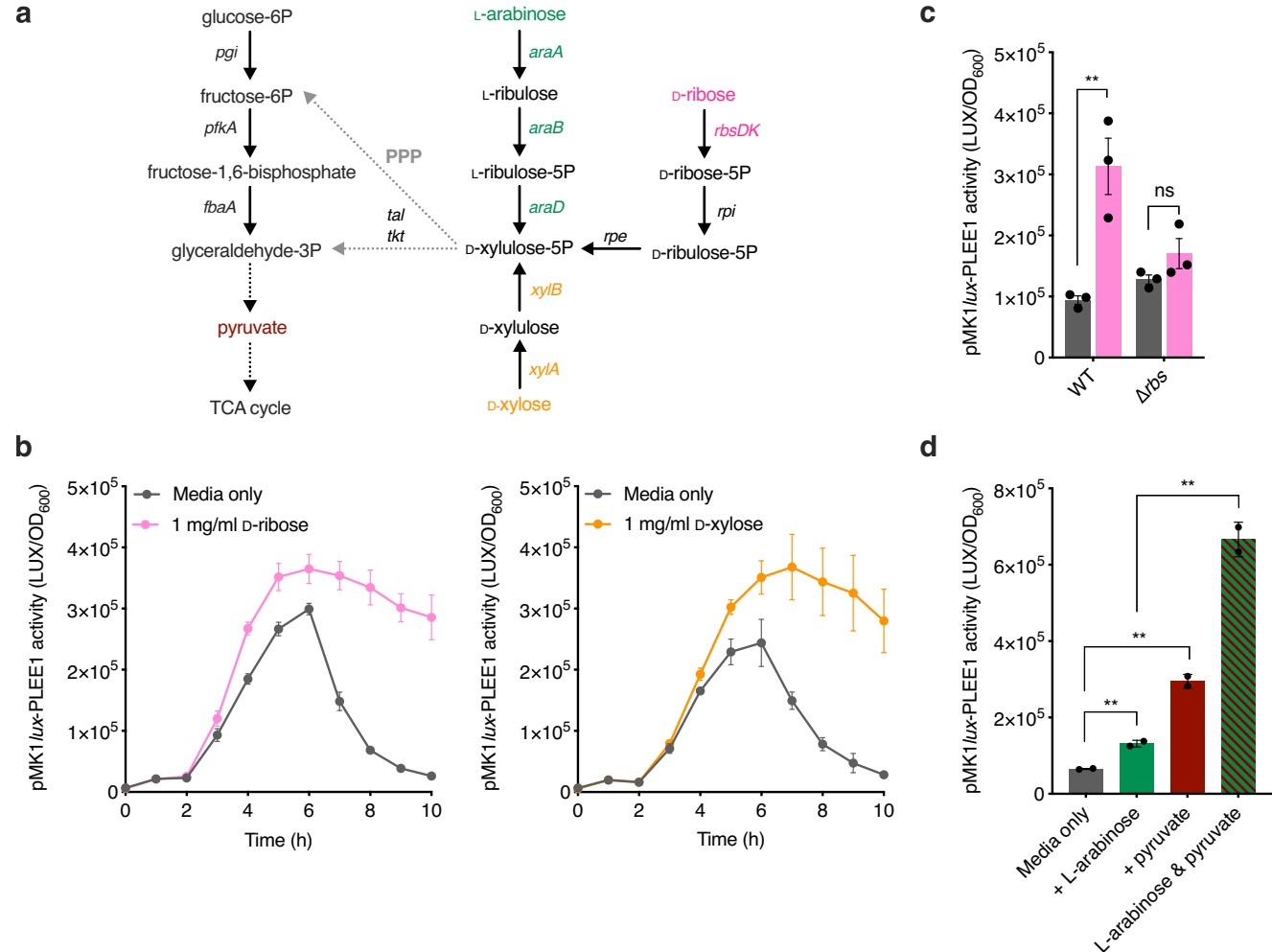

**Fig. 6 | A generalised mechanism for T3SS regulation by pentose sugar metabolism. a** Schematic illustrating where the L-arabinose, D-xylose and D-ribose metabolic pathways converge with glycolysis (left-hand side) via the Pentose Phosphate Pathway (PPP). The genes involved in each conversion are illustrated. Single steps are depicted as a solid arrow. Multiple steps are abbreviated to dotted arrows. **b** Transcriptional reporter assay of EHEC transformed with pMK1*lux*-PLEE1 cultured in MEM-HEPES alone or supplemented with 1 mg/ml of D-ribose (pink) or D-xylose (orange). Data are depicted as luminescence units (LUX) divided by the optical density ($OD_{600}$) of the culture at each timepoint. **c** pMK1*lux*-PLEE1 reporter assay of WT EHEC and the Δ*rbs* mutant cultured in MEM-HEPES alone (grey) or supplemented with D-ribose (pink). **d** pMK1*lux*-PLEE1 reporter assay of WT EHEC cultured in MEM-HEPES alone or supplemented with 1 mg/ml L-arabinose (green), 0.2% pyruvate (magenta) or a mixture of both (dashed). For (**c**, **d**), statistical significance was determined using a two-tailed student's *t*-test (significant $P = 0.0093$, 0.0031, 0.0035 and 0.0087 where indicated from left to right on **c**, **d**). Error bars represent the standard error of the mean ($n = 3$ biological replicates for (**c**); $n = 2$ biological replicates for **d**). ** and ns indicate $P < 0.01$ or not significant respectively for all graphs. Source data are provided in the Source Data file.

from the labs of Conway and Cohen describing carbon nutrition of *E. coli* in vivo has shown that L-arabinose is utilised by both commensal and EHEC strains in the streptomycin-treated mouse intestine[28]. Furthermore, they described how EHEC metabolises L-arabinose earlier in the hierarchy of carbon utilisation than a commensal strain when grown on a mixture of sugars. This suggests that L-arabinose is an important nutrient for *E. coli* to sustain growth within the mammalian intestine and that EHEC has evolved an enhanced ability to do this.

An enhanced ability of EHEC to utilise L-arabinose may be achieved by more efficient breakdown within or uptake into the cell. Our discovery of the novel L-arabinose ABC transporter Aau suggests that the latter may be particularly important in providing EHEC with a competitive edge in nature. Indeed, despite Aau being encoded on an O-island without a dedicated regulatory system[15], we have found that its activation relies on the core genome-encoded AraC in a manner akin to the canonical L-arabinose utilisation machinery[36]. This results in a rapid and coordinated response to the sugar by inducing the expression of multiple uptake systems. While we are currently investigating

the underlying mechanisms (affinity and kinetics of Aau uptake in comparison to the canonical L-arabinose uptake systems) of this transporter, one could speculate that encoding multiple systems with differing affinities for their substrate might provide a greater ability for scavenging scarce nutrients in the dynamic and highly competitive environment of the gut. Additionally, there have been other reports of Aau induction in ecologically relevant scenarios that support its likely role in EHEC host interaction. Transcriptome studies of EHEC in response to spinach and lettuce lysates found that the canonical L-arabinose genes and those encoding Aau were significantly upregulated, indicating a potential dietary source of L-arabinose[29,47]. Furthermore, the Aau genes (alongside the canonical L-arabinose systems) were found to be significantly upregulated during growth in bovine digestive contents. Finally, the permease was identified as being one of the most highly expressed EHEC proteins during human infection using in vivo antigen detection technology[31], which is particularly noteworthy considering that *C. rodentium* (a dedicated murine pathogen) does not encode this system[48]. This therefore provides

compelling evidence that Aau, and its role in ʟ-arabinose scavenging, likely represents a crucial element of the real-world ruminant reservoir or human infectious context.

Our discovery that ʟ-arabinose metabolism is required for regulation of the T3SS reveals insights into how we perceive virulence factor control in the host context. Traditionally, nutrients or small metabolites are thought to be "sensed" either in the cytosol post-uptake by a cognate transcription factor, or from the environment by periplasmic interactions with two-component systems. This is indeed true of pyruvate, which has been described as a virulence-inducing signal in many species beyond *E. coli*. For example, both *Staphylococcus aureus* and *Salmonella* Typhimurium upregulate virulence gene expression in response to host-derived pyruvate via two-component systems, the latter of which is particularly noteworthy here as this regulation occurred independently of any growth advantage[49,50]. Our proposed mechanism provides evidence that the endogenous metabolism of certain molecules, that do not in themselves act as "signals", can cause dramatic shifts in the pathogen response. This follows on from our recent study where we found that metabolism of microbiota-derived 1,2-propanediol generates the short-chain fatty acid propionate, which in turn provides a signal that activates LEE expression in *C. rodentium* independently of a growth advantage[21]. In a comparable manner, uropathogenic *E. coli* produces excess pyruvate via serine deamination, which in turn results in increased sensing of pyruvate via a two-component system and subsequent uptake[51]. These studies collectively point to alternative mechanisms of virulence and fitness regulation whereby "signals" are generated endogenously by the pathogen based on the biochemical status of the environment. Importantly, it also alludes to a more complex view of the consequences of niche adaptation, whereby metabolic systems can provide additional underappreciated benefits to a pathogen through the regulation of genetically unlinked virulence factors. The mechanism of ʟ-arabinose induced LEE expression also reinforces the benefit of being able to dynamically shift between available nutrients in vivo that are likely to be in constant flux and thus would aid in limiting any associated repressive effects of catabolite repression on virulence gene expression[22,27,28].

Finally, while the canonical ʟ-arabinose utilisation pathway has been understood and exploited in bacterial genetics for decades, the relevance of ʟ-arabinose metabolism in promoting pathogenesis has only recently emerged. Dietary ʟ-arabinose utilisation by *S.* Typhimurium promotes expansion in the gastrointestinal tract of mice[52]. This occurs via a mechanism involving an intrinsic alpha-*N*-arabinofuranosidase that liberates the sugar from dietary polysaccharides. Dietary fibre is traditionally thought to provide a benefit in maintaining colonisation resistance via the maintenance of microbiota composition, the mucosal barrier and short-chain fatty acid production[53–55]. The data from Ruddle et al. and ours therefore highlight the diverse strategies that enteric pathogens can also use to take advantage of fibre-derived sources of nutrition. Indeed, EHEC utilises several strategies to achieve this. For example, sensing pectin-derived galacturonic acid promotes initial expansion in the mouse gut while also directly regulating LEE expression throughout the infection lifespan[56]. These studies collectively highlight the diverse strategies that enteric pathogens use to exploit the host diet and aid in overcoming colonisation resistance.

In summary, we have identified a mechanism by which A/E enteric pathogens regulate an essential virulence factor in response to host-associated ʟ-arabinose and identified novel genes involved in its uptake from the environment. Our data reveal that important pathogenic processes can be controlled via the central metabolism of sugars independently of any effects on growth, warranting a different perspective on this process and its potential impact in bacterial pathogens. We anticipate that by expanding our view of how nutrient "sensing" occurs via metabolism we have the potential to discover

additional mechanisms of bacterial virulence regulation that are highly relevant to the host context. While our studies using *C. rodentium* cannot rule out that ʟ-arabinose metabolism may benefit EHEC on a nutritional level in certain contexts (such as during bovine colonisation or human infection respectively), the results highlight how virulence factor regulation can provide additional benefits for overcoming colonisation resistance. Lastly, by identifying further functional distinctions between EHEC and *C. rodentium* (exemplified here by the EHEC-specific Aau), we may reveal genetic factors that dictate host-range of these human and murine pathogens exclusively.

## Methods

### Ethics statement
All animal experiments were performed in strict accordance with the United Kingdom Home Office Animals Scientific Procedures Act of 1986 under the personal project licence number PP8850146 (protocols 1 and 2). The experiments were subject to local ethical approval and consideration given to the refine, reduce and replace principals wherever possible so that all efforts were made to minimize animal suffering.

### Bacterial growth conditions
All bacterial strains used are listed in Supplementary Table 1. Single colonies were used to inoculate 5 mL LB media before overnight culture at 37 °C and back dilution into LB or MEM-HEPES (T3SS-inducing conditions) to an $OD_{600}$ of 0.05 for specific assays or growth analysis. For sole carbon source experiments, overnight cultures were first washed three times in PBS to remove trace LB and used to inoculate M9 minimal media supplemented with a single carbon source (ᴅ-glucose, ʟ-arabinose, ᴅ-ribose or ᴅ-xylose). Sugars were added to media at concentrations of 0.5 or 1 mg/ml (3.45 and 6.9 mM) where indicated. All experiments were performed at 37 °C with shaking at 200 rpm. When necessary, antibiotics were used at the following concentrations: 100 μg/mL ampicillin, 50 μg/mL kanamycin, 20 μg/mL chloramphenicol, 100 μg/mL streptomycin and 50 μg/mL nalidixic acid. All chemicals and media were purchased from Merck or Thermo Fisher Scientific.

### Lambda red recombineering
Non-polar single-gene deletions were generated using the Lambda Red recombineering system[57]. Briefly, the FRT-Kanamycin or FRT-Chloramphenicol cassette was amplified from pKD4 or pKD3 respectively, using primers containing 50 bp overhangs homologous to the regions directly up- and downstream regions of the gene of interest. Parental strains carrying pKD46 were cultured in SOB media (Ampicillin; 30 °C) containing 10 mM ʟ-arabinose to an $OD_{600}$ of 0.4 before electroporation with 100 ng of the PCR product described above. Subsequent recovery was carried out in SOC media at 37 °C prior to plating the transformations on LB-agar containing the appropriate antibiotic for the selection of recombinants. Positive mutants were identified by colony PCR and verified by Sanger sequencing. To remove the resistance cassette, positive mutants were subsequently transformed with pCP20, plated on LB-agar containing Ampicillin and grown overnight at 30 °C to induce FLP recombinase activity. Colonies were re-streaked non-selectively at 42 °C to cure the strains of pCP20. The plasmids used are listed in Supplementary Table 2. All primers used are listed in Supplementary Table 3.

### Plasmid construction
All complementation and reporter plasmids were generated by standard restriction enzyme cloning except for pSU-*araBADE* and pSU-*aau*, which were generated using the NEBuilder® HiFi DNA Assembly Cloning kit as per the manufacturer's instructions. All plasmids generated are listed in Supplementary Table 2 and all primer sequences are listed in Supplementary Table 3. For reporter plasmids, promoter

regions comprising of approximately 300 bp upstream of the gene of interest were PCR amplified from genomic DNA using primers containing EcoRI (5′) and BamHI (3′) restriction sites. PCR products were gel extracted, digested, phosphatase treated and ligated into pMK1*lux*. This resulted in a transcriptional reporter whereby the respective promoter for a gene of interest was fused to the *luxCDABE* operon from *Photorhabdus luminescens*. For complementation constructs, the gene/operon of interest was PCR amplified from genomic DNA using primers containing BamHI (5′) and XbaI (3′) restriction sites. PCR products were ligated into pSU-PROM or pACYC184. All restriction enzymes, Q5 high fidelity polymerase, Antarctic Phosphatase and T4 ligase were purchased from New England Biolabs. Cloning was confirmed by sequencing of plasmid inserts.

### Animal experiments

Strains of interest were used to mono- and co-infect BALB/c mice (Charles River) as previously described[19]. In brief, groups of five adult female mice (7–8 weeks old) were inoculated by oral gavage using 200 μL of PBS suspension containing $2 \times 10^9$ CFU of either a single strain or two strains mixed at a ratio of 1:1. For the analysis of bacterial colonisation, stool samples were aseptically recovered, weighed, and homogenised in PBS before being serially diluted. The number of CFU per gram of stool was determined by plating onto LB-agar with the appropriate antibiotic. Experiments were typically performed on two independent occasions and CFU counts were analysed using the Mann–Whitney *U*-test. For experiments involving *E. coli*, streptomycin-resistant strains encoding an intact Aau locus were used and drinking water was supplemented with streptomycin (5 g/L) two days prior to oral gavage and maintained for the duration of the experiment[28]. For experiments involving *C. rodentium*, animals were given normal drinking water.

### LUX-promoter fusion reporter assays

Promoter activity was determined by measuring cell density ($OD_{600}$) and absolute luminescence of cultures carrying LUX reporter fusions using a FLUOstar Omega microplate reader (BMG Labtech). Relative luminescence units were calculated by dividing absolute luminescence values by the $OD_{600}$ at each given timepoint. Assays were performed as a time-course in 200 μL wells of white-walled, clear flat-bottom, 96-well polystyrene microtiter plates or taken as single measurements. Alternatively, endpoint luminescence measurements were taken from reporter strains grown in MEM-HEPES for 2 h before being spiked with the indicated sugar and grown for a further 5 h. Control cultures were spiked with the equivalent volume of sterile PBS. Individual culture volumes were inoculated with the strain of interest at a ratio of 1:100.

### Secreted protein SDS-PAGE profiling

T3SS-mediated protein secretion was profiled by culturing EHEC in 50 mL MEM-HEPES at 37 °C to late exponential phase ($OD_{600}$ of 0.8–0.9). The cell-free supernatant was separated from the cellular fraction by centrifugation and filtration using a 0.4 μm filter. The total secreted protein was precipitated from the cell-free supernatant with 10% ice-cold trichloroacetic acid at 4 °C overnight. Secreted proteins were concentrated by centrifugation for 1 h at $4000 \times g$. The supernatant was discarded, and pellets were resuspended in 150 μL of 1× lithium dodecyl sulphate buffer and boiled at 95 °C for 10 min. Samples were normalised by $OD_{600}$ at the point of harvesting. All protein samples were separated by SDS-PAGE using a 4–12% Bis-Tris NuPAGE gel (Invitrogen) and running at 150 V for 90 min, before staining with Coomassie blue.

### Immunoblot analysis

Secreted protein fractions and corresponding whole-cell lysates (obtained by boiling the cell pellet in sample buffer as above) were used for immunoblot analysis. Samples were separated by SDS-PAGE

and were transferred from a 4–12% Bis-Tris NuPAGE gel to a 0.45 μM nitrocellulose membrane (GE Healthcare) using an XCell II Blot module (Invitrogen) at 30 V for 90 min. Membranes were then blocked with 5% skim milk powder in PBS-Tween at room temperature for one hour before being incubated with primary antibodies for one hour. Membranes were washed three times in PBS-Tween for 10 min each before incubating for one hour with secondary antibodies. Primary antibodies used were anti-EspD (1:2500) and anti-GroEL (1:25,000). Secondary antibodies used were anti-mouse and anti-rabbit HRP-conjugated (1:20,000). Immunoblots were incubated with SuperSignal West Pico chemiluminescent substrate (Pierce) for five minutes before imaging using a G:Box Chemi system (Syngene).

### HeLa cell adhesion assays and fluorescence microscopy

For adhesion assays, HeLa cells were seeded onto sterile coverslips coated with rat tail collagen ($10^4$ cells per coverslip in 12-well plates) in DMEM supplemented with 10% foetal calf serum and 1% Penicillin/Streptomycin. Cells were incubated overnight at 37 °C with 5% $CO_2$. Prior to infection, cells were washed twice with PBS and fresh MEM-HEPES supplemented with or without 1 mg/mL L-arabinose was added. A 40 μL volume of MEM-HEPES containing EHEC grown to $OD_{600}$ 0.9 and back-diluted to 0.1 was added to each coverslip. Plates were centrifuged at 400 rpm for 3 min and incubated at 37 °C with 5% $CO_2$ for two hours. Wells were then washed with fresh media and incubated for a further 3 h. The wells were washed three times and fixed with 4% paraformaldehyde for 15 min at room temperature. Wells were washed an additional two times and permeabilised with 0.1% Triton X-100 for 5 min. The wells were washed twice more before incubation with ActinRed™ 555 ReadyProbes™ reagent (Rhodamine phalloidin; Invitrogen). Tissues were washed for a final time and mounted onto glass slides using Fluoroshield™ with DAPI. All washes were done using sterile PBS. For all assays, EHEC transformed with an *rpsM*:GFP reporter plasmid was used. Slides were imaged using a Zeiss Axioimager at ×40 magnification. Data were collected by imaging 20 random fields of view across three coverslips prepared on independent occasions.

### RNA extraction

EHEC cultures were mixed with 2 volumes of RNAprotect reagent (Qiagen), incubated at room temperature for 5 min and harvested by centrifugation. Total RNA extraction was done using a Monarch® Total RNA Miniprep Kit (New England Biolabs) prior to treatment with TURBO DNAse (Ambion). DNA-free RNA samples were analysed by Qubit (ThermoFisher Scientific) and assessed for degradation using agarose gel electrophoresis.

### Quantitative real-time PCR (RT-qPCR)

RNA samples were normalised, and cDNA synthesis was performed using a LunaScript RT SuperMix kit (New England Biolabs). RT-qPCR was performed on the resulting cDNA using a LightCycler 96 Real-Time PCR system (Roche) and Luna Universal qPCR Master Mix kit (New England Biolabs). Reactions were performed as technical replicates within each of the three biological replicates. All genes were normalised against the housekeeping gene, *groEL*. All primers used in RT-qPCR were checked for efficiency (90–110%) using standards made from template gDNA (100, 20, 4, 0.8 and 0.16 ng/μl). The data was analysed by the $2^{-\Delta\Delta CT}$ method[58]. Primer sequences are listed in Supplementary Table 3.

### RNA-sequencing and transcriptome analysis

Ribosomal depletion and library assembly of DNA-free total RNA samples were carried out using an Illumina Ribo-Zero Tru-seq kit according to the manufacturer's specifications. Samples were processed by the Newcastle Genomics Core Facility. Sequencing was carried out using a mid-range run on the Illumina Next-seq platform, generating 75-bp single-end reads. Raw read quality was checked

using FastQC (https://www.bioinformatics.babraham.ac.uk/projects/fastqc/). To estimate transcript abundance, SALMON was used under the default parameters for the mapping of reads to the *E. coli* O157:H7 strain EDL933 reference sequence (Accession GCA_000006665), retrieved from Ensembl. Transcript level counts outputted from SALMON were then summarised at the gene level using tximport[59]. DESeq2 (v.1.28.1) was used to normalise RNA-seq count data and identify differentially expressed genes between conditions[60]. Differentially expressed genes displayed an absolute fold change of >1.5 and had a false discovery rate of 5% (adjusted $P < 0.05$). Volcano plots were generated in R studio using the enhanced volcano package (v.1.6.0). Functional enrichment and gene ontology analysis were performed using STRING to identify protein-protein interactions related to the differentially expressed genes identified using default parameters and excluding any disconnected nodes[61]. Sequencing data has been deposited in the NCBI Gene Expression Omnibus under the accession number GSE262155.

### L-arabinose quantification

L-arabinose was quantified from homogenised and filtered luminal and colon tissue samples of BALB/c mice using an L-arabinose/D-galactose assay kit (Megazyme) as per the manufacturer's instructions.

### Thin-layer chromatography (TLC)

Cell-free supernatants of EHEC cultured in MEM-HEPES alone or supplemented with L-arabinose were prepared by removing 1 mL of culture every hour, removing the supernatant by centrifugation and passing through a 0.2 μM filter. Using TLC aluminium plates (Merck), 6 μL of sample for each time point was spotted 1 cm from the bottom and allowed to dry. The plates were run in solvent (1-butanol:acetic acid:water at a ratio of 2:1:1), dried and sugars visualised by immersion in Orcinol stain.

### Gene carriage analysis

To assess the frequency of carriage of the Z0415-9 locus and LEE island across the *E. coli* lineage, paired-end sequence read data for 1067 strains previously described[25] were retrieved from the National Centre for Biotechnology Information (NCBI) Sequence Read Archive (accessed 16th September 2021), using the 'prefetch' and 'fastq-dump' tools within the SRA Toolkit v2.9.0-mac64 (http://ncbi.github.io/sra-tools). FastQC v0.11.8 (http://www.bioinformatics.babraham.ac.uk/projects/fastqc/) was used to generate quality statistics for the paired-end reads, which were aggregated into a single report and visualised using MultiQC v1.11[62]. Kraken v2.0.8-beta was used to screen the raw Illumina sequencing data for contamination against the NCBI RefSeq Database[63]. Raw reads were filtered using Trimmomatic v0.36 by removing low-quality bases and read pairs together with Illumina adaptor sequences (settings: LEADING:3, TRAILING:3, MINLEN:36, SLIDINGWINDOW:4:15)[64]. The complete chromosomes of the 56 genomes were used to simulate error-free reads. This was done using the software package ART (version ART-MountRainier-2016-06-05) which simulated paired-end Illumina reads to 60× coverage with an insert size of 340 ± 40 bp[65]. The average sequence coverage depth was estimated using the Burrows–Wheeler Aligner (BWA) v0.7.15[66]; SAMtools v1.2[67]; Picard v2.7.1 (https://github.com/broadinstitute/picard); the Genome Analysis Tool Kit v3.2-2 (GATK)[68,69]; BEDTools v2.18.2[70]; and SNPEff v4.1 as implemented in SPANDx v3.2[71,72]. In brief, the quality-trimmed paired-end reads were mapped to the complete chromosome of strain EDL933 (GenBank: AE005174)[15]. Trimmed reads for the draft genomes were de novo assembled using Shovill v1.0.4 (https://github.com/tseemann/shovill) (--gsize 5 M, --minlen 200, --mincov 10, --opts "--sc"), which implements: Seqtk v1.3-r106 (https://github.com/lh3/seqtk); Lighter v1.1.2[73]; FLASH v1.2.11[74]; SPAdes v3.13.1[75]; Samclip v0.2 (https://github.com/tseemann/samclip); SAMtools v1.8[67]; BWA-MEM v0.7.17-r1188[66]; and Pilon v1.22[76]. QUAST v4.5 was used to assess the de

novo assembly metrics generated from Shovill by comparing each draft assembly to strain EDL933[77]. We identified and excluded the sequence data for certain strains from further analysis based on sequencing coverage being below 20-fold, the presence of contaminants, and the genome length from de novo assembly falling outside the upper (Q3) and lower (Q1) 1.5× interquartile range (i.e., <4,095,491 bp or >5,889,907 bp) ($n = 157$). To assess the publicly available strains for the presence of strain mixtures, paired-end reads were mapped onto the chromosome of EDL933 using SPANDx to generate annotated single-nucleotide polymorphisms (SNPs) and insertions and deletions (INDELs) matrices. Heterozygous SNPs in each genome were identified from GATK UnifiedGenotyper VCF output. MLST v2.19.0 (https://github.com/tseemann/mlst) with default settings was used to characterise the multi-locus sequence type (MLST) of the remaining 949 strains ($n = 948$ excluding EDL933) against the *E. coli* MLST allelic profiles hosted on PubMLST[78]. High-resolution analysis of genetic variants was performed using SPANDx with EDL933 as a reference[72]. A maximum likelihood phylogenetic tree from the non-recombinant SNP alignment was generated using RAxML v8.2.10 (GTR-GAMMA correction) through the optimisation of 10 distinct, randomized maximum parsimony trees[79]. The resulting phylogenetic tree was visualised using FigTree v1.4.4 (http://tree.bio.ed.ac.uk/software/figtree/).

### Protein bioinformatics

AlphaFold modelling was performed using AlphaFold version 2.1.1 implemented on the Monash University MASSIVE M3 computing cluster[80,81]. The amino acid sequences of the periplasmic binding protein (Z0415) (minus the predicted signal peptide), ATPase (Z0416-7) and two permease subunits (Z0418/Z0419) of Aau were provided and modelling was run in multimer mode, with a single molecule of each subunit requested. The quality of five ranked models produced by AlphaFold was assessed based on the pLDDT score and compared for consistency with the top-ranked model used for further analysis and figure generation.

To identify ABC transporters closely related to Z0415-19, a phylogenetic analysis was conducted based on the associated amino acid sequence. The ABC_tran_pfam domain (pfam00005) was searched against EHEC EDL933 using the IMG/M server 'find function' tool. The returned hits were then filtered based on the predicted substrates to select only those predicted to transport sugar substrates. The amino acid sequences were then exported and aligned in MEGAX (V 10.1.8) using MUSCLE[82]. The evolutionary history was inferred by using the Maximum Likelihood method and Le_Gascuel_2008 model. The tree with the highest log likelihood (−11697.05) is shown. The percentage of trees in which the associated taxa clustered together is shown next to the branches. Initial tree(s) for the heuristic search were obtained automatically by applying Neighbor-Join and BioNJ algorithms to a matrix of pairwise distances estimated using the JTT model and then selecting the topology with superior log likelihood value. A discrete Gamma distribution was used to model evolutionary rate differences among sites (5 categories (+$G$, parameter = 1.6636)). The tree is drawn to scale, with branch lengths measured in the number of substitutions per site. This analysis involved 12 amino acid sequences. There were a total of 546 positions in the final dataset. Evolutionary analyses were conducted in MEGA X[83].

### Statistical analyses

Graphs were generated and statistical analyses were performed using GraphPad Prism version 8. Test details for each experiment can be found within the associated figure legends. The student's *t*-test was used for comparing two groups with normally distributed data and equal variance. The Mann–Whitney *U*-test was used to compare two groups that were not normally distributed and had unequal variance (mono-infections). The Wilcoxon signed-rank test was used to

compare matched samples (co-infections or co-colonisations). Fisher's exact test was used in the analysis of Odd's ratio contingency tables. *P*-values of less than or equal to 0.05 were considered statistically significant.

## Reporting summary

Further information on research design is available in the Nature Portfolio Reporting Summary linked to this article.

## Data availability

The transcriptomic sequencing data is available from the NCBI Gene Expression Omnibus under the accession number GSE262155. The source data for Figs. 2–6, Supplementary Figs. 3–5 and 7–12 are provided as a Source Data file. Publicly available genome sequences were obtained from the NCBI Sequence Read Archive. Source data are provided with this paper.

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

## Acknowledgements
We are very grateful to Dr Nicky O'Boyle, Dr Kate Beckham and Dr Kesha Josts for their critical appraisal of the work. We also would like to thank Professor Janet Quinn for microscope access, the Comparative Biology Centre staff for their support with the animal studies and the Newcastle University Genomics Core Facility for RNA-sequencing. This work was supported by a Springboard Award from the Academy of Medical Sciences/Wellcome Trust [SBF005\1029], a Royal Society Research Grant [RGS\R2\202100], a Medical Research Council Career Development Award [MR/X007197/1] and a Faculty Fellowship (Newcastle University) awarded to J.P.R.C. E.C.L was supported by a Wellcome Trust Discovery Award [226644/Z/22/Z]. C.C. was supported by a PhD studentship from the Barbour Foundation.

## Author contributions
C.C. and J.P.R.C conceptualised and designed the research. C.C., R.T.W. and J.P.R.C. performed the research. C.C., R.T.W., L.C.B., R.G. and J.P.R.C. analysed the data. C.J.S., S.A.B., R.G., E.C.L. and J.P.R.C. contributed reagents/analytical tools. C.C. and J.P.R.C. wrote the paper with input from all other authors.

## Competing interests
The authors declare no competing interests.
