## [Peer Review File · Nature Communications]

Metabolism of L-arabinose converges with virulence regulation to promote enteric pathogen fitnessREVIEWER COMMENTS

Reviewer #1 (Remarks to the Author):

In this interesting manuscript the authors present various lines of evidence to suggest a new role for pentose sugar metabolism in regulation of virulence factor expression in two enteric pathogens. While not shown beyond doubt, the suggestion is that this is via the intracellular metabolite pyruvate, which is produced during the catabolism of these sugars. The evidence for induction of virulence factors by L-arabinose in particular is studied extensively and the lack of its metabolism results in loss of this phenotype, which is then confirmed in vivo in *Citrobacter* model of infection. While this part of the manuscript is strong, it is prefaced by work on the identification of a 'novel' L-arabinose transporter in EHEC, which is a weaker part of the paper and in fact raises important questions that are not answered. The authors should consider the necessity for this section of the paper and how it fits into the overall and convincing story about pentose sugar metabolism.

Specifically then, in the first results section the authors focus on their discovery of an uncharacterised ABC transporter in EHEC (Z0415-9), which they argue correlates with the presence of the LEE.

The data presented in Fig. 1b is very interesting and show the Aau genes present in diverse *E. coli* phylotypes, including D and F, B2, E and B1. The authors make a statement about the correlation of Aau with LEE carriage, but this conclusion seems to be largely driven by the over-representation of the LEE +ve phylotype E (which are genetically very similar to each other as evidenced from the tree). I would imagine that if each of the phylotypes were represented equally then this would not be the case. I think the authors can be confident in saying that the ST11 LEE +ve strains they pretty much always have the Aau genes.

As the data for Fig 1C is captured in the text and I don't think Fig 1A needs to be in the main paper (there is nothing surprising here in this prediction - it looks like a regular ABC transporter), then I would suggest that Fig. 1b now becomes Fig. 1 in toto which would make it larger on the page and easier to interpret.

When the authors start the experimental work (line 119) they don't tell us what EHEC strain they use (having just shown us in Fig. 1b that they are diverse). This must be stated and I am assuming that they have used an ST11 strain to build on their finding from Fig. 1b.

The authors then show expression of the genes are induced by L-Ara and that this requires AraC, and hence conclude they are embedded within the normal host regulatory network.

In Fig. 2b the data is quite hard to interpret, both as the two colours are quite similar but also that circles are used in every case such that it is hard to distinguish the data types near zero. This could be split into two plots, one with no ara and one with. You could replace the growth data from Part a (put it in extended data) and make part b clearer to the reader (it is very nice data).

The authors then show nicely that deletion of the aau genes had no effect on growth with L-ara (Extended Data Fig. 3), while there was a clear difference in competitiveness in vivo when these genes are disrupted (see later comment).

To learn more about the aau genes, the authors then do some nice RNA seq and find araFGH, araE and araBAD as well as their aau genes being strongly upregulated by L-Ara.

They should comment on other L-ara transporters that are also upregulated in this data (Supp Table 1), for example the araJ system is 12-fold induced and the gaf (ytf) system is also significantly induced. This is another L-arabinose transporter (see Drousiotis et al 2023 doi: 10.1099/mic.0.001308) and work from the Wade lab in 2014 also showed that these genes were under AraC control. (see Stringer et al 2014 doi: 10.1128/JB.01007-13).

While the data nicely provides additional information on the AraC regulon in EHEC, the link to LEE is less clear to me (Fig 3B). The authors state that several LEE genes were significantly upregulated and looking at the data in the graphic in Fig. 3B I can see escF as being the most

induced, but this gene is not present in Supp Table 1? Why is this?

I could map the *escT* and map genes but both are only 1.04 and 1.06 log₂ induced, which is very close to the noise in the data.

Have I missed something? Overall, I would like to see some more analysis that suggests that the LEE genes are more generally upregulated than other sets of gene from their data.

Regardless of this the LEE story is followed up and data is presented showing upregulation of these genes by L-ara. The authors show us in Fig. 3b that there are 5 LEE sets of genes, but when they do the reporter gene fusion in Fig. 3C it is not clear which promoter they are measuring? Can this be clarified please.

It would be useful for the authors to report the L-ara concentrations in $\mu\text{M}/\text{mM}$ in addition to mg/ml to help understand the concentrations used and how this related to the affinities of the transporter and enzymes being used. Various other data then strongly supports the idea that L-Ara induces LEE.

The authors then show that LEE induction is lost in an *araC* mutant, which they acknowledge is a complicated phenotype as loss of both transporter and metabolic gene expression.

The authors then investigate LEE induction in strains lacking individually 2 of the many known L-ara transporters, but surprisingly (to me) don't include their own Aau system in this experiment? Why was this not included? This seems very surprising given in their first results section they show aau is important for colonisation and as a reader I am wanting to know what the mechanism behind this is?

As the authors state, it is not a surprise that there is not a total loss of phenotype with the individual transporter mutants, although their finding that *araE* is most important is not surprising given the concentration of L-ara being used for growth is in the mM range (this is an assumption of 1 mg/ml being used as it is not stated in the legend to Fig. 4 when L-ara concentration is added).

The data in Fig. 4c is very nice, but again shows that AraE alone can support the induction of the L-ara response.

One wonders if a much lower L-ara concentration was used if AraFGH/GafABCD or Aau would be important and perhaps in vivo it is use of Aau (as a very high affinity transporter) that might explain its in vivo phenotype?

Hence, this section is a little unsatisfactory as one is unsure of what the conclusion is about the function of aau and transport more generally. Specifically, if there are multiple transporters then one would predict little effect of knocking them out individually, which is seen in the work on the ones tested, however, aau, which is not tested in the same way, gives a colonisation defect in vivo, suggesting that its function cannot be easily substituted by other transporter.

The authors then continue the paper on much stronger ground. They show that other pentose sugars can also induce LEE like L-Ara. This they argue is consistent with pyruvate production being the metabolic trigger for LEE induction, which is known already and the shown again here. This appears a sensible hypothesis, but earlier (line 64) it is stated that glucose represses LEE, however, glucose metabolism would also produce pyruvate - how do the authors reconcile these data?

Finally using *Citrobacter* the authors show the the L-Ara effect is mediated via expression of LEE thereby pulling the threads of the L-Ara story together.

Overall, the paper provides important new evidence for the use of dietary derived sugars as cues for virulence factor expression by enteric pathogens. The authors might wish to consider how best to use the aau related material because at the moment it is difficult to reconcile the in vivo phenotype in EHEC (Fig. 2) with this being solely due to a defect in L-ara uptake. This is both because a direct role for Aau in L-ara transporter is not demonstrated in the paper and its effect on LEE induction is not tested in Fig. 4, while evidence for *araE* and *araGFH* is presented and found to not abolish LEE induction.

Reviewer #2 (Remarks to the Author):

The manuscript is well written and centres around the link between sugar metabolism and virulence factor regulation in *E. coli* O157, a zoonotic bacterial pathogen. I consider the impacts demonstrated are modest but they highlight the subtle and integrated nature of the regulation of a major colonisation factor, the type 3 secretion system of *E. coli* O157. The authors show that that L-arabinose and similar sugars can increase expression of the secretion system indirectly, likely by an increase in pyruvate level in the bacteria. The authors have shaped a satisfying 'story' with a series of elegant in vitro and in vivo assays and models that allow the L-arabinose effect to be measured. My main criticism is that in places, I consider the 'sell' is a bit strong and suggest some areas for consideration or re-phrasing. I think the Discussion is excellent and really ties the work together as it brings in elements of other work that strengthen the overall argument.

Abstract

L28. "arabinose induces expression..." perhaps semantic but the effects shown are very subtle compared to other regulatory inputs into T3S and also the marked variation observed in T3S between *E. coli* O157 isolates. If you induce 'a birth' it makes it happen, in this case the sugar indirectly does help promote expression as another part of a complex set of regulatory inputs, but its not IPTG. You use 'enhances' on L179 which seems more appropriate. Also the molecule actually 'enhancing' is not L-arabinose, making the statement incorrect on several levels.....

L34. "and highlights the unexpected impact that nutrient utilisation can have in enteric pathogens." I think 'unexpected' is a stretch. For example, (as discussion) there is a large body of work in enteric pathogens showing how important catabolite repression by glucose is for virulence factor expression, Lrp with leucine and alanine and the feast famine response, etc....

Introduction

Overall, there seems to be a missing aspect on where *E. coli* O157 spends the majority of its time (in the ruminant host) and therefore the main environment that is shaping its evolution and nutrient requirement. With that established, then a key question is whether variation in this type of regulation could be important for likelihood and severity of incidental human infection.

Results

L115. 'This suggests that there may be conflating evolutionary pressures for LEE-encoding EHEC strains to acquire Z0415-9,' It is an interesting analysis of the genes across 900+ *E. coli*, but I don't know if its possible to imply a direction of acquisition? It may be the case that the LEE locus and Stx phages were acquired after this region in strains that were EHEC ancestors? Certainly, the analysis does imply the transporter has not been retained or selected in certain *E. coli* backgrounds over others and it does correlate well with LEE+ EHEC strains. Would you expect the same association with LEE negative (and non Stx) serogroups in cattle that can become EHEC, showing that it is more about the general strain background and host environment rather than the link to the LEE per se?

L246 '.....and demonstrated that growth in the presence of both exogenous pyruvate and L-arabinose displayed an additive effect'. Does this result make sense, would you not expect pyruvate levels be tightly regulated so that sugar breakdown would only be necessary as pyruvate levels fall. How does pyruvate increase T3S and if that is published then as your prediction and results then isn't any compound that feeds through to pyruvate levels likely to have the same phenotype.....in which case is 'L-arabinose' the main story? I suppose the finding of the additional uptake system is the connection although how this actually benefits yet has not been shown.

L272 The final experiments with CR in mice are interesting but more of the primary data should be shown. First of all, as with the STEC expts in the streptomycin-treated mice, the araBCD KO is not complemented, but I appreciate that this is difficult and probably would need to be in the chromosome and ideally re-instatement at the wild type locus. Again, ideally, the CR experiments could be done with a diet with and without added arabinose to really show its arabinose in the animal. I think the fixed Ler induction is clever, but it is possible that virtually any change in the strain that left it less successful in the mouse model might well benefit from this change in terms of colonisation? Although, conversely, you might expect such fixed regulation to be a negative

compared to the wild type? Only the CI is shown between the two backgrounds that both have the fixed ler expression with the statement that the wild type and C. rodentiumPler-const displayed similar colonisation dynamics (L277). I consider that the actual data needs to be provided to see how the levels of colonisation are impacted between the different strains, not just CIs. These are logical experiments but in my opinion they don't rule out other inputs and this could be discussed.

Discussion

L296. '....suggesting that EHEC has an enhanced ability to scavenge L-arabinose in the gut and maximise its competitiveness through the convergence of virulence and metabolism.' Appreciate you have used the term 'suggesting' here, but the work on Aau does raise the very obvious question of why it would be an advantage over the canonical system? They are regulated apparently in the same way, so what would be the advantage of the second system? Does it enable the bacteria to scavenge L-arabinose at lower levels? You do cover this aspect from L315+ but it is down as further work and does leave the reader hanging.....Also in this section you have the citation that links though to the abstract statement around high expression in human infection (ref 45). This is unusual as you are expecting to read this in a section in the results. This result though does make my earlier comment about variation in capacity to cause human infection even more relevant and there is a lot of interest in defining why specific STEC are more of a threat to human health. I look forward to reading about progress in this area in the future.

David Gally

REVIEWER COMMENTS

We would like to thank both reviewers for their positive comments and interest in our work. We value the constructive feedback and have tried our best to address all the points raised. Please note any references made to Figure numbers correspond to the revised version of our manuscript.

Reviewer #1 (Remarks to the Author):

In this interesting manuscript the authors present various lines of evidence to suggest a new role for pentose sugar metabolism in regulation of virulence factor expression in two enteric pathogens. While not shown beyond doubt, the suggestion is that this is via the intracellular metabolite pyruvate, which is produced during the catabolism of these sugars. The evidence for induction of virulence factors by L-arabinose in particular is studied extensively and the lack of its metabolism results in loss of this phenotype, which is then confirmed in vivo in *Citrobacter* model of infection. While this part of the manuscript is strong, it is prefaced by work on the identification of a 'novel' L-arabinose transporter in EHEC, which is a weaker part of the paper and in fact raises important questions that are not answered. The authors should consider the necessity for this section of the paper and how it fits into the overall and convincing story about pentose sugar metabolism.

We accept the criticism but still feel that it is important to report this section in our paper, seeing as this was the discovery that prompted us to investigate pentose sugar uptake and its impact on EHEC virulence regulation in the first instance (also highlighted by reviewer 2). As detailed in the following responses, we have now clarified all the points raised by the reviewer to our best ability and provided new experimental evidence for the role of Aau that we believe strengthens the manuscript and its narrative.

Specifically then, in the first results section the authors focus on their discovery of an uncharacterised ABC transporter in EHEC (Z0415-9), which they argue correlates with the presence of the LEE.

The data presented in Fig. 1b is very interesting and show the Aau genes present in diverse *E. coli* phylotypes, including D and F, B2, E and B1. The authors make a statement about the correlation of Aau with LEE carriage, but this conclusion seems to be largely driven by the over-representation of the LEE +ve phylotype E (which are genetically very similar to each other as evidenced from the tree). I would imagine that if each of the phylotypes were represented equally then this would not be the case. I think the authors can be confident in saying that the ST11 LEE +ve strains they pretty much always have the Aau genes.

This is a valid point with respect to the evolutionary structure of phylogroup E, however our strain cohort was not intentionally selected for this reason. We employed a strategy similar to our previously published work on presence/absence of specific genetic loci in *E. coli* (PMID: 25526369) in order to capture the diversity of the species phylogeny as accurately and broadly as possible. As can be seen, individual phylogroups are not overrepresented per se in terms of total numbers within the analysis. Regardless, we are glad that the reviewer agrees that we can

confidently assume that LEE+ve strains are strongly correlated with Aau carriage.

As the data for Fig 1C is captured in the text and I don't think Fig 1A needs to be in the main paper (there is nothing surprising here in this prediction - it looks like a regular ABC transporter), then I would suggest that Fig. 1b now becomes Fig. 1 in toto which would make it larger on the page and easier to interpret.

We have incorporated this change. Additionally, we have removed the phrase 'novel' from the abstract with reference to Aau to tone down the discovery of an uncharacterised, albeit well studied family of, transporter.

When the authors start the experimental work (line 119) they don't tell us what EHEC strain they use (having just shown us in Fig. 1b that they are diverse). This must be stated and I am assuming that they have used an ST11 strain to build on their finding from Fig. 1b.

The strains used were all listed in the Supplementary information, but we agree it was not clear enough in the main text. For the Aau characterisation, we used the prototypical EHEC ST11 strain ZAP193. We chose this strain for the growth and in vivo experiments because it encodes an intact Aau locus. During our bioinformatic analysis, we noticed that our other commonly employed prototype strain (TUV93-0) contains a SNP in the ATPase component, resulting in a premature stop codon at position 259 (C>T). While not impacting on regulation or transcription of the locus, which is conserved in both strains, this results in a truncated and presumably non-functional ATPase (due to interruption of the Q-loop domain) and therefore could not be reliably used to test growth related phenotypes. We have now improved the text to clearly state this rationale:

Lines 144-146: "To study the role of Aau in EHEC growth and fitness, we employed strain ZAP193 (ST11) encoding a functionally intact Aau locus, due to TUV93-0 containing a SNP in the ATPase component of the transporter that results in a premature stop codon at position 259."

The authors then show expression of the genes are induced by L-Ara and that this requires AraC, and hence conclude they are embedded within the normal host regulatory network.

In Fig. 2b the data is quite hard to interpret, both as the two colours are quite similar but also that circles are used in every case such that it is hard to distinguish the data types near zero. This could be split into two plots, one with no ara and one with. You could replace the growth data from Part a (put it in extended data) and make part b clearer to the reader (it is very nice data).

We have incorporated this useful suggestion.

The authors then show nicely that deletion of the aau genes had no effect on growth with L-ara (Extended Data Fig. 3), while there was a clear difference in competitiveness in vivo when these genes are disrupted (see later comment).

Determining the functional role of Aau without an *in vitro* phenotype during growth on L-arabinose (likely due to the presence of multiple additional uptake systems) and therefore linking this system to the *in vivo* fitness defect was an obvious gap in our study. To provide more convincing evidence for the role of Aau in L-arabinose utilisation specifically, we have taken advantage of the TUV93-0 strain (encoding a transcribed but non-functional locus). We cloned the entire Aau locus from ZAP193 into pSUPROM, where it is under constitutive control of the strong Tat promoter. Introduction of this plasmid into wild type TUV93-0 did not enhance growth (presumably due to the activity of the canonical transport system/s). However, when introduced into the *araE* mutant background (which has a severe growth defect on L-arabinose), Aau expression results in partial recovery of growth on L-arabinose as a sole carbon source. These new data have now been included (Supplementary Figure 5) and support the hypothesis that Aau plays a biological role in the transport, and subsequent utilisation of L-arabinose as a substrate.

Lines 146-156: "Deletion of Aau in ZAP193 did not result in a significant defect during *in vitro* growth on L-arabinose as a sole carbon source, likely due to presence of the canonical transporters AraE or AraFGH (Supplementary Fig. 4). As an alternative way of assessing if Aau could potentially provide a benefit to EHEC, we cloned the entire locus from ZAP193 into plasmid pSUPROM (pSU-*aau*) where it is under constitutive control of the Tat promoter. Growth of TUV93-0 was first compared to $\Delta araE$, a mutant which displays a major fitness defect on L-arabinose as the sole carbon source (Supplementary Fig. 5a/b). However, when pSU-*aau* was introduced into the $\Delta araE$ background we observed that growth was partially recovered (Supplementary Fig. 5c). This suggests that Aau has the capacity to enhance L-arabinose uptake in combination with existing transporters and thus potentially increase fitness."

To learn more about the *aau* genes, the authors then do some nice RNA seq and find *araFGH*, *araE* and *araBAD* as well as their *aau* genes being strongly upregulated by L-Ara.

They should comment on other L-ara transporters that are also upregulated in this data (Supp Table 1), for example the *araJ* system is 12-fold induced and the *gaf* (*ytf*) system is also significantly induced. This is another L-arabinose transporter (see Drousiotis et al 2023 doi: 10.1099/mic.0.001308) and work from the Wade lab in 2014 also showed that these genes were under AraC control. (see Stringer et al 2014 doi: 10.1128/JB.01007-13).

This information has now been stated in the manuscript.

While the data nicely provides additional information on the AraC regulon in EHEC, the link to LEE is less clear to me (Fig 3B). The authors state that several LEE genes were significantly upregulated and looking at the data in the graphic in Fig. 3B I can see *escF* as being the most induced, but this gene is not present in Supp Table 1? Why is this?

I could map the *escT* and map genes but both are only 1.04 and 1.06 log₂ induced, which is very close to the noise in the data.

Have I missed something? Overall, I would like to see some more analysis that

suggests that the LEE genes are more generally upregulated than other sets of gene from their data.

To clarify this point, we identified upregulation (~ 2-fold absolute FC) across most of the LEE island (as illustrated by the heatmap), but only a number of these reached statistical significance (*espL*, *espG*, *map*, *escT*), hence why the rest are missing from the DEG list which summarises significance changes only. While ~2-fold increase might seem mild, this is not uncommon for the LEE considering that its basal expression levels are quite strong in MEM-HEPES without additional inducers and the fact that a multitude of inputs converge on its regulation, and therefore is not considered noise. The reason for the moderate increase was likely due to the time of the RNA sampling (after ~5 hours of growth in MEM-HEPES). When we performed the RNA-seq analysis we did not yet understand the dynamics of this regulatory event (i.e. the LEE phenotype increasing in strength at later timepoints in growth, when the switch to L-arabinose metabolism occurs). Our subsequent temporal reporter analysis highlighted this and allowed us to measure the effects on the LEE more accurately in subsequent assays. We have now included new RT-qPCR data, which validates this and shows clearly that the enhanced LEE1-5 expression is initially weak and increases across all operons later in the growth cycle (Supplementary Figure 7). Having tested LEE induction through three distinct methods, we are confident in the robustness of this observation. We have modified the text accordingly:

Lines 181-186: “While we anticipated shifts in expression of genes related to metabolism, we also noticed that majority of the LEE pathogenicity island displayed increased expression, with several genes (*escL*, *escT*, *espG* and *map*) reaching statistical significance (Fig. 3b). Additionally, the non-LEE encoded effectors *nleF* and *nleG6-3* were also significantly upregulated in response to L-arabinose. This data suggests that exposure to L-arabinose may influence the expression of virulence genes that are not related to pentose utilisation.”

Lines 198-200: “RT-qPCR analysis confirmed a significant increase in transcription across all five LEE operons only at this later stage of growth, thus explaining the mild increase observed in our RNA-seq analysis.”

Regardless of this the LEE story is followed up and data is presented showing upregulation of these genes by L-ara. The authors show us in Fig. 3b that there are 5 LEE sets of genes, but when they do the reporter gene fusion in Fig. 3C it is not clear which promoter they are measuring? Can this be clarified please.

The reporter measures transcription from the LEE1 promoter. LEE1 encodes the master regulator of the LEE (*Orf1* or *Ier*), which is known to coordinate expression of LEE1-5. Therefore, it is commonly employed in the field to use LEE1 expression as a proxy for T3SS activity. We have now modified the text to more clearly explain this for readers that are less familiar with the EHEC field:

Lines 188-195: “The LEE is essential for EHEC pathogenicity and is responsive to gut-associated cues. Its associated genes are encoded largely across five polycistronic operons (LEE1-5) and are co-regulated by the

activity of the master regulator Ler (encoded by the first open reading frame of LEE1)¹². We therefore reasoned that L-arabinose may provide a fitness advantage through regulating T3SS activity either in concert to, or independent from a role in host-associated nutritional competition. To assess the dynamics of LEE induction by L-arabinose, we engineered a transcriptional reporter of the LEE1 promoter, used as a proxy to measure T3SS expression (EHEC transformed with pMK1/*LUX-P_{LEE1}*)."

It would be useful for the authors to report the L-ara concentrations in uM/mM in addition to mg/ml to help understand the concentrations used and how this related to the affinities of the transporter and enzymes being used. Various other data then strongly supports the idea that L-Ara induces LEE.

This has now been included.

The authors then show that LEE induction is lost in an *araC* mutant, which they acknowledge is a complicated phenotype as loss of both transporter and metabolic gene expression.

The authors then investigate LEE induction in strains lacking individually 2 of the many known L-ara transporters, but surprisingly (to me) don't include their own *Aau* system in this experiment? Why was this not included? This seems very surprising given in their first results section they show *aau* is important for colonisation and as a reader I am wanting to know what the mechanism behind this is?

The aim of this experiment was to narrow down the mechanism of LEE regulation by L-arabinose. As such, we needed to separate the effects of breakdown, uptake into the cell or direct regulation by *AraC* to achieve this. Therefore, the *Aau* mutant was not included here simply because we wanted to use a transporter mutant with a measurable growth phenotype and established role in arabinose uptake (*AraE*). As could be seen, *AraE* appears to be the main transporter, at least in vitro, and as such its deletion results in a dramatically reduced level of enhanced LEE expression. Therefore, by focusing on *AraE* in this experiment, simply as a tool to study uptake, we could confidently conclude that the LEE phenotype was related to the action of *AraBAD*, and not *AraC* or the presence of L-arabinose in the cytosol via transport. It's important to reiterate that the point of this experiment was not to measure the effects of individual transporters on LEE regulation. We don't suggest that there is a direct functional link between the LEE and *Aau* (or any individual transporter for that matter). Rather, we are suggesting that the addition of an extra transport system provides potentially greater capacity to uptake a sugar that would therefore have downstream benefits to cell in terms of T3SS activation. However, the reviewer is right that this is a logical thing to test, and we have now included this experiment in the revised manuscript (Supplementary Figure 9). As can be seen, deletion of *Aau* only results in a very minor reduction to LEE expression compared to the wild type under these conditions, suggesting that the system likely contributes only partially to LEE regulation in response to L-arabinose, at least in vitro (as would be expected given the presence of multiple transporters and apparent importance of *AraE* in vitro).

Lines 229-231: “In addition to this, we tested LEE expression in the Δaau background showing that, at least in vitro, this system does not significantly mediate the enhanced LEE phenotype.”

As the authors state, it is not a surprise that there is not a total loss of phenotype with the individual transporter mutants, although their finding that *araE* is most important is not surprising given the concentration of L-ara being used for growth is in the mM range (this is an assumption of 1 mg/ml being used as it is not stated in the legend to Fig. 4 when L-ara concentration is added).

When optimising our experiments, we tested all our growth and LEE reporter assays in different mutant backgrounds across a range of L-arabinose concentrations and did not detect any relative contribution of individual transporters or phenotypic distinctions depending on which concentration was used.

The data in Fig. 4c is very nice, but again shows that AraE alone can support the induction of the L-ara response.

One wonders if a much lower L-ara concentration was used if AraFGH/GafABCD or Aau would be important and perhaps in vivo it is use of Aau (as a very high affinity transporter) that might explain its in vivo phenotype?

The reviewer is correct in suggesting that different affinities of the transport systems may reflect a discreet role in vivo role, when sugars are likely scarce. However, this would also be complicated by many other factors that would be difficult to control for, such as the dynamics of local sugar availability in the gut niche and differing expression levels/activities of each transport system in vivo. As mentioned in our discussion, we are currently working towards an in-depth biochemical study investigating the comparative affinities/kinetics of arabinose transport more generally in EHEC, which we still feel is a level of analysis beyond the scope of our current paper and would not affect its overall conclusions.

However, specifically in relation to Fig 4c, we would like to reiterate our point above that the aim of this experiment was not to determine the individual contribution of each transporter on the LEE regulation phenotype. It was a synthetic scenario designed to test the relative contribution of transport generally, metabolism or sensing via AraC on LEE expression. The results did indeed successfully separate transport and regulation by AraC from the role of metabolism (AraBAD) in the regulatory effect on the LEE, which was what we aimed to achieve.

Hence, this section is a little unsatisfactory as one is unsure of what the conclusion is about the function of *aau* and transport more generally. Specifically, if there are multiple transporters then one would predict little effect of knocking them out individually, which is seen in the work on the ones tested, however, *aau*, which is not tested in the same way, gives a colonisation defect in vivo, suggesting that its function cannot be easily substituted by other transporter.

We hope that by clarifying our rationale for Figure 4 (previous points), alongside our new data on Aau function, we have provided more convincing evidence supporting its potential contribution to arabinose uptake. Regarding the in vivo phenotype, it's also important to reiterate that the Streptomycin mouse model used for EHEC is not

a model for pathogenesis (as we specifically highlight in the text and is well known in the field). Therefore, the fitness defect associated with Aau deletion may well reflect a T3SS-independent phenotype in the context of this particular model. As *Citrobacter* does not encode the Aau system, we could not specifically test its role in pathogenesis using an in vivo system.

The authors then continue the paper on much stronger ground. They show that other pentose sugars can also induce LEE like L-Ara. This they argue is consistent with pyruvate production being the metabolic trigger for LEE induction, which is known already and the shown again here. This appears a sensible hypothesis, but earlier (line 64) it is stated that glucose represses LEE, however, glucose metabolism would also produce pyruvate - how do the authors reconcile these data?

Yes, it is correct that multiple pathways generating pyruvate could be interpreted as beneficial for LEE expression. We hypothesise that other nutrient sources encountered in nature may be more beneficial in this scenario likely due to the known repressive effects of glucose on dampening maximal LEE expression (as has been shown previously and cited by us). We therefore propose that switching to a different nutrient source, such as pentose sugars, may be a strategy for EHEC to avoid this negative side effect on LEE expression (as mentioned in our original discussion).

Finally using *Citrobacter* the authors show the the L-Ara effect is mediated via expression of LEE thereby pulling the threads of the L-Ara story together.

Overall, the paper provides important new evidence for the use of dietary derived sugars as cues for virulence factor expression by enteric pathogens. The authors might wish to consider how best to use the aau related material because at the moment it is difficult to reconcile the in vivo phenotype in EHEC (Fig. 2) with this being solely due to a defect in L-ara uptake. This is both because a direct role for Aau in L-ara transporter is not demonstrated in the paper and its effect on LEE induction is not tested in Fig. 4, while evidence for araE and araGFH is presented and found to not abolish LEE induction.

We thank the reviewer for their supportive comments. We hope that our responses above and additional data included on the function and specificity of Aau provides a more convincing link into our studies related to L-arabinose regulation of the LEE and the potential contribution of this new transport system to EHEC fitness.

Reviewer #2 (Remarks to the Author):

The manuscript is well written and centres around the link between sugar metabolism and virulence factor regulation in *E. coli* O157, a zoonotic bacterial pathogen. I consider the impacts demonstrated are modest but they highlight the subtle and integrated nature of the regulation of a major colonisation factor, the type 3 secretion system of *E. coli* O157. The authors show that that L-arabinose and similar sugars can increase expression of the secretion system indirectly, likely by an

increase in pyruvate level in the bacteria. The authors have shaped a satisfying 'story' with a series of elegant in vitro and in vivo assays and models that allow the L-arabinose effect to be measured. My main criticism is that in places, I consider the 'sell' is a bit strong and suggest some areas for consideration or re-phrasing. I think the Discussion is excellent and really ties the work together as it brings in elements of other work that strengthen the overall argument.

We thank the reviewer for their supportive comments and for highlighting the complex nature of LEE regulation. We have toned down our discovery as suggested and improved the text where indicated to contextualise our findings better.

Abstract

L28. "arabinose induces expression..." perhaps semantic but the effects shown are very subtle compared to other regulatory inputs into T3S and also the marked variation observed in T3S between E. coli O157 isolates. If you induce 'a birth' it makes it happen, in this case the sugar indirectly does help promote expression as another part of a complex set of regulatory inputs, but its not IPTG. You use 'enhances' on L179 which seems more appropriate. Also the molecule actually 'enhancing' is not L-arabinose, making the statement incorrect on several levels.....

It is not uncommon for regulatory inputs into the LEE to be subtle (see similar point in response to Reviewer 1 above). This system is responsive to a huge number of signalling events and transcription factors (see our reviews on this topic PMID: 34224961 PMID: 30559275 PMID: 26097473). However, for accuracy, we have removed "induced" and modified the text in the abstract to more generally capture the ultimate point that it is pentose metabolism (not specifically arabinose) that mediates our phenotype.

Lines 23-36: "Virulence and metabolism are often interlinked to control the expression of essential colonisation factors in response to host-associated signals. Here, we identified a new transporter of the dietary monosaccharide L-arabinose that is widely encoded by the zoonotic pathogen enterohaemorrhagic *Escherichia coli* (EHEC), required for full competitive fitness in the mouse gut and highly expressed during human infection. Discovery of this transporter suggested that EHEC strains have an enhanced ability to scavenge L-arabinose and therefore prompted us to investigate the impact of L-arabinose on pathogenesis. Accordingly, we discovered that L-arabinose enhances expression of the EHEC type 3 secretion system, increasing its ability to colonise host cells, and that the underlying mechanism is dependent on products of its catabolism rather than the sensing of L-arabinose as a signal. Furthermore, using the murine pathogen *Citrobacter rodentium*, we show that L-arabinose metabolism provides a fitness benefit during infection via virulence factor regulation, as opposed to supporting pathogen growth. Finally, we show that this mechanism is not restricted to L-arabinose and extends to other pentose sugars with a similar metabolic fate."

L34. "and highlights the unexpected impact that nutrient utilisation can have in enteric pathogens." I think 'unexpected' is a stretch. For example, (as discussion) there is a large body of work in enteric pathogens showing how important catabolite

repression by glucose is for virulence factor expression, Lrp with leucine and alanine and the feast famine response, etc....

We have toned down the closing statement in the abstract to focus more on the conclusion of our own findings.

Lines 36-38: "This work highlights the importance integrating central metabolism with virulence regulation in order to maximise competitive fitness of enteric pathogens within the host-niche."

Introduction

Overall, there seems to be a missing aspect on where *E. coli* O157 spends the majority of its time (in the ruminant host) and therefore the main environment that is shaping its evolution and nutrient requirement. With that established, then a key question is whether variation in this type of regulation could be important for likelihood and severity of incidental human infection.

This is a valid point. Of course, L-arabinose is present in animals, not just humans (we state this in our manuscript), and therefore this system (as well as the canonical L-arabinose utilisation machinery) probably provides an advantage in several contexts. However, our study was not aimed at differentiating between the two hosts and we therefore focused the introduction on the mechanisms of EHEC pathogenesis, seeing as the bulk of our data centres around T3SS regulation via nutrient utilisation (which we do not state is strictly relevant to humans). However, we agree with the reviewer that this aspect should be incorporated more clearly, and we have now expanded the introduction to capture this point.

Lines 49-52: "Enterohaemorrhagic *Escherichia coli* (EHEC) is a zoonotic pathogen that is carried asymptotically by ruminant mammals and transmitted to human hosts typically via contaminated meat and fresh produce. In humans, EHEC causes severe diarrhoeal illness and, in extreme cases, renal failure^{6,7}."

Furthermore, we have also found evidence that the *Aau* genes are upregulated during growth in bovine digestive content (PMID: 30352567) and have incorporated this information into the manuscript to broaden the potential relevance to zoonoses, and not strictly human infection.

Lines 367-369: "Furthermore, the *Aau* genes (alongside the canonical L-arabinose systems) were found to be significantly upregulated during growth in bovine digestive contents."

Results

L115. 'This suggests that there may be conflating evolutionary pressures for LEE-encoding EHEC strains to acquire Z0415-9,' It is an interesting analysis of the genes across 900+ *E. coli*, but I don't know if its possible to imply a direction of acquisition? It may be the case that the LEE locus and Stx phages were acquired after this region in strains that were EHEC ancestors? Certainly, the analysis does imply the transporter has not been retained or selected in certain *E. coli* backgrounds over others and it does correlate well with LEE+ EHEC strains. Would you expect the

same association with LEE negative (and non Stx) serogroups in cattle that can become EHEC, showing that it is more about the general strain background and host environment rather than the link to the LEE per se?

Yes, we agree that this is likely more reflective of the strain background and host environment. We don't propose a chronology of acquisition across different *E. coli* phylogroups. Neither do we suggest a direct functional link between the two systems. We are arguing that a greater capacity to scavenge a scarce nutrient potentially provides a benefit in terms of the downstream regulatory effects on the LEE (or nutrition depending on the context). However, we agree that "conflating evolutionary pressure" could be misleading and have modified the phrasing in this section to more accurately capture this point.

Lines 125-128: "This suggests that there may be a related evolutionary pressure for LEE-encoding EHEC strains to acquire or retain Z0415-9, implicating its associated function in pentose sugar scavenging as being potentially beneficial for EHEC infection."

L246 '.....and demonstrated that growth in the presence of both exogenous pyruvate and L-arabinose displayed an additive effect'. Does this result make sense, would you not expect pyruvate levels be tightly regulated so that sugar breakdown would only be necessary as pyruvate levels fall. How does pyruvate increase T3S and if that is published then as your prediction and results then isn't any compound that feeds through to pyruvate levels likely to have the same phenotype.....in which case is 'L-arabinose' the main story? I suppose the finding of the additional uptake system is the connection although how this actually benefits yet has not been shown.

As explained above in the response to reviewer 1, we believe the switch to pentose metabolism provides a benefit by producing pyruvate from an alternative nutrient source and as a result stimulates LEE expression, without the known negative effects of glucose on this process (as discussed in our paper). Thus, the additive effect does make sense as arabinose utilisation is regulated by glucose catabolite repression, rather than levels of pyruvate as a signal. We do not know the mechanism of how pyruvate stimulates the LEE. In an attempt to address this, we generated a mutant in the pyruvate repressor protein (PdhR) as a logical starting point but the absence of this regulator did not reverse the phenotype.

Nevertheless, the reviewer is correct in stating that generation of pyruvate as a way to promote LEE regulation is therefore not restricted to arabinose. We agree that the result generalises our findings more towards pentose sugars more broadly and not specifically L-arabinose (our revised abstract captures this more clearly and we have reordered figures 5 and 6 accordingly to reflect this). However, this is still an L-arabinose story given our discovery of Aau and investigation of L-arabinose metabolism/transport as the model system to study LEE regulation in this context.

L272 The final experiments with CR in mice are interesting but more of the primary data should be shown. First of all, as with the STEC expts in the streptomycin-treated mice, the araBCD KO is not complemented, but I appreciate that this is difficult and probably would need to be in the chromosome and ideally re-instatement

at the wild type locus. Again, ideally, the CR experiments could be done with a diet with and without added arabinose to really show its arabinose in the animal. I think the fixed Ler induction is clever, but it is possible that virtually any change in the strain that left it less successful in the mouse model might well benefit from this change in terms of colonisation? Although, conversely, you might expect such fixed regulation to be a negative compared to the wild type? Only the CI is shown between the two backgrounds that both have the fixed ler expression with the statement that the wild type and *C. rodentium* Pler-const displayed similar colonisation dynamics (L277). I consider that the actual data needs to be provided to see how the levels of colonisation are impacted between the different strains, not just CIs. These are logical experiments but in my opinion they don't rule out other inputs and this could be discussed.

Our additional data on the role of Aau now more clearly links its function to L-arabinose specifically, and therefore we feel now supports the in vivo data better. Dietary supplementation with L-arabinose would not necessarily be a more direct method of achieving this, given that there are multiple transport systems at play. Furthermore, we (and others cited in our paper) have detected L-arabinose in the intestinal contents of mice. Therefore, it is already present in a natural capacity and, as such, dietary modification represents a different question that we feel is not needed to support the data we have shown here. Mutations in *araBAD* and their effects on in vivo fitness have been previously demonstrated in multiple other model systems (as cited in our paper) thus we are confident in its role in host-pathogen interactions. As such, our constitutive Ler experiment more accurately supports the conclusions of our paper and complements the in vivo phenotype in itself. It's a valid query that this system may result in an ability to overcome several fitness defects (it would be impossible to test everything) but we don't believe this is the case. The mutation does not alter the overall colonisation dynamics of *Citrobacter* compared to the wild type, and presence of the microbiota results in clearance at the expected experimental stage. Additionally, it has been shown by the Frankel lab that antibiotic treatment using this strain results in relocation from the epithelium of the colon to the lumen (PMID: 29262319). Thus, this strain is still susceptible to external inputs and does not render it capable of colonising the colon indefinitely. We have used this strain previously where we showed that the colonisation dynamics match the parental strain (PMID: 30305622), hence why we didn't include it here in the original submission. However, for clarity, we have now included this data in the revised version as suggested by the reviewer (Supplementary Figure 12).

Discussion

L296. '....suggesting that EHEC has an enhanced ability to scavenge L-arabinose in the gut and maximise its competitiveness through the convergence of virulence and metabolism.' Appreciate you have used the term 'suggesting' here, but the work on Aau does raise the very obvious question of why it would be an advantage over the canonical system? They are regulated apparently in the same way, so what would be the advantage of the second system? Does it enable the bacteria to scavenge L-arabinose at lower levels? You do cover this aspect from L315+ but it is down as further work and does leave the reader hanging.....

We appreciate the criticism. However, the same logic could be applied to any one of the several existing L-arabinose transport systems in *E. coli*. We are not suggesting that this is the first instance of a bacterium encoding more than one transport system for a common nutrient and we feel it is logical to suggest that presence of an additional system would provide a competitive advantage over strains that do not encode this. Of course, differing affinities/kinetics of each transporter might explain their relative roles but (as pointed out above) this would also be complicated by several other factors, such as differing expression levels of each system specifically in the in vivo setting. As such, we suggested this line of enquiry as future work because it does not affect our overall conclusions. We feel a study of that kind would require direct biochemical comparison of each transporter system from the same strain in tandem, representing a large independent body of work and as such is beyond the scope of the current study. We have modified the text in this discussion point to more clearly highlight this hypothesis:

Lines 359-363: “While we are currently investigating the underlying mechanisms (affinity and kinetics of Aau uptake in comparison to the canonical L-arabinose uptake systems) of this transporter, one could speculate that encoding multiple systems with differing affinities for their substrate might provide a greater ability for scavenging scarce nutrients in the dynamic and highly competitive environment of the gut.”

Also in this section you have the citation that links though to the abstract statement around high expression in human infection (ref 45). This is unusual as you are expecting to read this in a section in the results. This result though does make my earlier comment about variation in capacity to cause human infection even more relevant and there is a lot of interest in defining why specific STEC are more of a threat to human health.

There was no reason for this other than the fact that it was not data generated by us and thus not one of our results. However, to make the point clearer we have now mentioned this link earlier in the results section and cited the appropriate literature for the reader.

I look forward to reading about progress in this area in the future.

David Gally

Many thanks for the encouraging and supportive feedback.

REVIEWERS' COMMENTS

Reviewer #1 (Remarks to the Author):

The authors provide a strong and convincing rebuttal to both reviewers including various bits of additional data. Importantly this addresses one of the gaps in the first version of the paper, which was the lack of any direct evidence that the Aau system was an L-arabinose transporter, which is now demonstrated using partial complementation of an araE mutant strain. The authors have clarified and added other bits of analysis and toned down some of their terms, which overall makes the paper more complete

Reviewer #2 (Remarks to the Author):

The authors have made some changes and added data that helps address some of the issues raised. The differences in several of the assays are quite subtle but the overall subject area is important and the key result is the fine tuning of colonisation factor expression using basic metabolism as a signal of niche.

REVIEWERS' COMMENTS

Reviewer #1 (Remarks to the Author):

The authors provide a strong and convincing rebuttal to both reviewers including various bits of additional data. Importantly this addresses one of the gaps in the first version of the paper, which was the lack of any direct evidence that the Aau system was an L-arabinose transporter, which is now demonstrated using partial complementation of an araE mutant strain. The authors have clarified and added other bits of analysis and toned down some of their terms, which overall makes the paper more complete

Reviewer #2 (Remarks to the Author):

The authors have made some changes and added data that helps address some of the issues raised. The differences in several of the assays are quite subtle but the overall subject area is important and the key result is the fine tuning of colonisation factor expression using basic metabolism as a signal of niche.

We thank both reviewers for their positive comments and support of our work. We feel that the constructive feedback has helped result in a much-improved manuscript.